# Snf1 and yeast GSK3-β activates Tda1 to suppress glucose starvation signaling

Kazuki Nonaka[1], Kohei Nishimura [ID][1 ✉], Kazuma Uesaka[2], Emi Mishiro-Sato [ID][3], Minako Fukase[1], Rei Kato[1], Fumihiko Okumura [ID][4], Kunio Nakatsukasa[5], Keisuke Obara [ID][1 ✉] & Takumi Kamura [ID][1 ✉]

## Abstract

In budding yeast, the presence of glucose, a preferred energy source, suppresses the expression of respiration-related genes through a process known as glucose repression. Conversely, under glucose starvation conditions, Snf1 phosphorylates and activates downstream factors, relieving this repression and allowing cells to adapt. Recently, the Tda1 protein kinase has been implicated in these glucose starvation responses, although its function remains largely uncharacterized. In this study, we demonstrate that Snf1 and yeast glycogen synthase kinase 3-beta (GSK3-β) independently phosphorylate and activate Tda1, which in turn phosphorylates Hxk2 at Ser15. The Ser483 and Thr484 residues of Tda1 are critical for its activation by Snf1, while the Ser509 residue is crucial for its activation by yeast GSK3-β. Importantly, under glucose starvation conditions, the TDA1 deletion mutant shows increased expression of respiration-related genes and a faster growth rate compared to wild-type cells, which is opposite to what is observed in SNF1 and yeast GSK3-β deletion mutants. These findings suggest that Tda1 is activated by Snf1 and yeast GSK3-β, and functions as a suppressor of the glucose starvation signaling.

**Keywords** Glucose Starvation; Glycogen Synthase Kinase 3-beta; Hxk2; Snf1; Tda1
**Subject Categories** Metabolism; Signal Transduction

## Introduction

Living organisms have evolved various response pathways, including starvation response pathways, to adapt constant environmental changes. ATP, often called the cell's energy currency, is essential for numerous life processes. All organisms have developed mechanisms to maintain ATP levels in response to environmental changes. In eukaryotic cells, the serine/threonine kinase 5' AMP-activated protein kinase (AMPK) serves as a key sensor of cellular energy status. AMPK is a heterotrimeric complex composed of α, β, and γ subunits (Davies et al, 1994). Under nutrient-deprived conditions, the γ subunit detects a drop in the cellular ATP/(AMP + ADP) ratio, leading to AMPK activation through the phosphorylation of a conserved threonine residue on the α subunit by AMPK kinases, such as LKB1 (Hardie et al, 2012; Hawley et al, 1996; Shaw et al, 2004; Woods et al, 2005; Xiao et al, 2007; Xiao et al, 2011). In *Saccharomyces cerevisiae*, the highly conserved yeast homolog of AMPK, Snf1, regulates metabolic shifts in response to changes in environmental glucose levels. When glucose is abundant, yeast cells primarily rely on glycolysis for energy, suppressing other metabolic pathways such as aerobic respiration, even when oxygen is present —a phenomenon known as glucose repression (Kayikci O and Nielsen 2015). However, when glucose levels drop, alternative metabolic pathways are activated. This metabolic shift is driven largely by widespread changes in gene expression, with Snf1 playing a central role. Upon glucose starvation, Snf1 is activated through phosphorylation at the Thr210 residue by upstream kinases like Tos3, Sak1, and Elm1 (Hong et al, 2003), while the inhibitory effects of SUMOylation and deprotonation of the N-terminal polyHis motif, which occur under high-glucose conditions, are also relieved (Simpson-Lavy et al, 2015; Simpson-Lavy and Johnston, 2013; Simpson-Lavy and Kupiec, 2022). Once activated, Snf1 promotes the expression of genes involved in aerobic respiration, β-oxidation, and alternative energy metabolism by phosphorylating downstream transcriptional activators such as Adr1, Cat8, and Sip4, as well as the transcriptional repressor Mig1 (RandezGil et al, 1997; Treitel et al, 1998; Vincent and Carlson, 1999; Young et al, 2002).

Hxk2, a hexokinase involved in glycolysis, catalyzes the conversion of glucose to glucose-6-phosphate (Lobo and Maitra, 1977). Interestingly, beyond its enzymatic function, Hxk2 also acts as a transcriptional repressor, working alongside Mig1 to regulate gene expression. In low-glucose conditions, activated Snf1 is thought to phosphorylate Hxk2 at Ser15, promoting its export from the nucleus and allowing gene expression to proceed (Ahuatzi et al, 2004; Ahuatzi et al, 2007; Fernández-García et al, 2012; Papamichos-Chronakis et al, 2004). However, a recent report has indicated that the nuclear translocation of Hxk2 can also occur under low-glucose conditions (Lesko et al, 2023). Moreover, other

[1]Graduate School of Science, Nagoya University, Furo-cho, Chikusa-ku, Nagoya 464-8602, Japan. [2]Graduate School of Bioagricultural Science, Nagoya University, Furo-cho, Chikusa-ku, Nagoya 464-8601, Japan. [3]Institute of Transformative Bio-Molecules (WPI-ITbM), Nagoya University, Furo-cho, Chikusa-ku, Nagoya 464-8601, Japan. [4]Department of Food and Health Sciences, International College of Arts and Sciences, Fukuoka Women's University, Fukuoka, Fukuoka 813-8529, Japan. [5]Graduate School of Science, Nagoya City University, Nagoya, Aichi 467-8501, Japan. ✉E-mail: Nishimura.kohei.x8@f.mail.nagoya-u.ac.jp; obara.keisuke.r2@f.mail.nagoya-u.ac.jp; kamura.takumi.k1@f.mail.nagoya-u.ac.jp

research has suggested that another protein kinase, Tda1, participates in Hxk2 phosphorylation (Kaps et al, 2015; Kettner et al, 2012). How Hxk2 is phosphorylated and regulates downstream gene expression is still controversial and requires further investigation. It has also been reported that Tda1, activated through Snf1-mediated phosphorylation at the Ser483 and Thr484 residues, phosphorylates histone H3 at Thr11 (Oh et al, 2020). Comprehensive studies have identified multiple phosphorylated serine and threonine residues on Tda1, hinting at the involvement of additional, yet unidentified, upstream kinases beyond Snf1 (Albuquerque et al, 2008; Chen et al, 2018; Holt et al, 2009; Lanz et al, 2021; MacGilvray et al, 2020; Soulard et al, 2010; Swaney et al, 2013; Zhou et al, 2021).

AMPK regulates cellular functions through interactions with various kinases, including glycogen synthase kinase 3-beta (GSK3-β). Both AMPK and GSK3-β share several substrates, such as glycogen synthase (Jorgensen et al, 2004). Previous studies suggest that these two kinases may regulate each other's activity through direct or indirect phosphorylation in humans and mice, indicating a close functional relationship (Suzuki et al, 2013). GSK3-β is inhibited by Akt and PKA, both of which are active under high-glucose conditions (Cross et al, 1995; Fang et al, 2000). Consequently, similar to AMPK, GSK3-β can be activated during glucose starvation when Akt and PKA are inactivated. GSK3-β's substrate phosphorylation requires prior 'priming phosphorylation' by other kinases in many eukaryotes, enabling it to tightly control multiple pathways, including those involved in nutrient starvation responses (Fiol et al, 1987). In Saccharomyces cerevisiae, there are four GSK3-β homologs: Rim11, Mck1, Mrk1, and Ygk3. The glucose starvation response mediated by yeast GSK3-β is thought to involve the stress-responsive transcription factors, Msn2 and Msn4, which share target genes with some of Snf1's downstream effectors (Hirata et al, 2003; Kuang et al, 2017). Interestingly, MSN2 and MSN4 have been identified as multicopy suppressors of SNF1 mutations, further emphasizing the functional connection between Snf1 and yeast GSK3-β (Estruch and Carlson, 1993).

In this study, we demonstrate that, in addition to Snf1, yeast GSK3-β also plays a role in regulating the phosphorylation of Hxk2 at serine 15 via Tda1 phosphorylation. Our results reveal that the Ser483 and Thr484 residues of Tda1 are important for activation by Snf1, while the Ser509 residue is crucial for activation by yeast GSK3-β, with all these residues being key for Hxk2 phosphorylation. Interestingly, the tda1Δ mutant shows increased expression of respiratory genes and enhanced growth under glucose starvation, in contrast to the snf1Δ and rim11Δ mck1Δ mutants. These findings suggest that Tda1, activated by both Snf1 and yeast GSK3-β, acts as a negative regulator in the glucose starvation signaling pathway.

# Results

## Tda1 phosphorylated by Snf1 directly phosphorylates Hxk2

It has been reported that both Snf1 and Tda1 act as protein kinases that directly phosphorylate Hxk2 at Ser15, although this remains controversial (Fernández-García et al, 2012; Kaps et al, 2015). To clarify this, we investigated whether Tda1 and Snf1 contribute to Hxk2 phosphorylation both in vivo and in vitro. First, we generated rabbit polyclonal antibodies specific to Hxk2 and Tda1, allowing us to detect the endogenous, untagged forms of these proteins via immunoblotting (Appendix Fig. S1A,B; Fig. 1A). Next, we examined the effect of Tda1 and Snf1 on Hxk2 phosphorylation at Ser15 under low-glucose conditions (Fig. 1A,B). Upon shifting the growth medium from high-glucose (2%) to low-glucose (0.05%), we observed the appearance of a higher molecular weight form of Hxk2 in wild-type (WT) cells, which was absent in the Hxk2-S15A mutant. Since Hxk2 was separated using a phos-tag gel, this shifted band corresponds to Hxk2 phosphorylated at Ser15. Notably, no Hxk2 phosphorylation was observed in the tda1Δ mutant and kinase-dead Tda1-K68R mutant (Appendix Fig. S2A,B), while the snf1Δ mutant still showed Hxk2 phosphorylation, albeit at a reduced level compared to WT cells. Additionally, under low-glucose conditions, WT cells exhibited an increase in higher molecular weight forms of Tda1, which were absent in the snf1Δ and kinase dead Snf1-K84R mutants (Fig. 1A,C; Appendix Fig. S2C,D). These slower-migrating Tda1 bands disappeared upon λ protein phosphatase treatment (Fig. 1D), indicating that they represent phosphorylated Tda1. We observed the interaction between Tda1 and Snf1 under both high- and low-glucose conditions in co-immunoprecipitation experiments (Fig. 1E). We then tested whether Snf1 phosphorylates Tda1 in vitro (Fig. 1F). Snf1, immunoprecipitated from WT cells grown in low-glucose medium, efficiently promoted Tda1 phosphorylation. In contrast, Snf1-K84R did not phosphorylate Tda1. To further confirm the roles of Tda1 and Snf1 in Hxk2 phosphorylation, we conducted in vitro phosphorylation assays using Hxk2 as the substrate (Fig. 1G). Tda1, immunoprecipitated from WT cells under low-glucose conditions, exhibited significant activity in phosphorylating Hxk2, whereas Tda1-K68R did not show such activity. (Fig. 1H). Tda1 from the snf1Δ mutant, which lacked the slower-migrating band, showed significantly reduced Hxk2 phosphorylation compared to WT-derived Tda1, though some activity was still detectable (Fig. 1G). Snf1 itself did not phosphorylate Hxk2, regardless of Tda1's presence. Tda1 successfully phosphorylated Hxk2-WT but failed to phosphorylate the Hxk2-S15A mutant (Fig. 1I). These findings demonstrate that under low-glucose conditions, Tda1 is phosphorylated and activated by Snf1, and this activated Tda1 phosphorylates Hxk2 at Ser15. Our observation that Snf1 binds to Tda1 even under high-glucose conditions (Fig. 1E) suggests that this interaction may suppress Tda1 activity in such conditions. In contrast, the loss of SNF1 partially activates Tda1, resulting in increased phosphorylation of Hxk2 compared to WT cells, as shown in Fig. 1A,B.

## The Ser483 and Thr484 residues of Tda1 are important for Hxk2 phosphorylation

Recent studies have shown that Tda1, activated by Snf1-mediated phosphorylation at the Ser483 and Thr484 residues, phosphorylates histone H3 at Thr11 (Oh et al, 2020). Our findings suggest that Snf1-dependent activation of Tda1 is also crucial for Hxk2 phosphorylation (Fig. 1A,B,G). To determine whether the Ser483 and Thr484 residues of Tda1 are necessary for this process, we examined Hxk2 phosphorylation levels in the Tda1-S483A/T484A mutant under low-glucose conditions (Fig. 2A,B). The mutant showed reduced Hxk2 phosphorylation compared to WT cells,

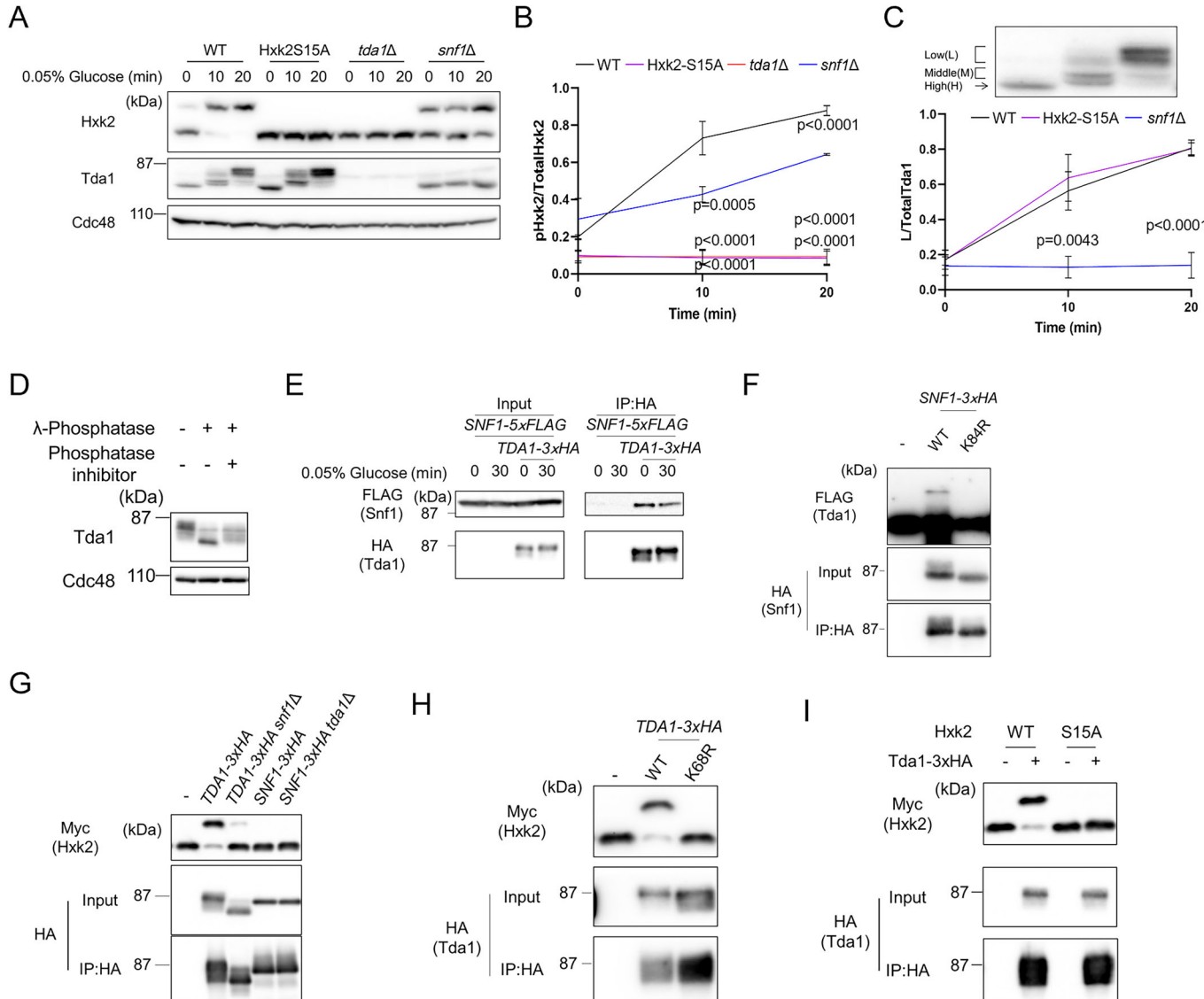

**Figure 1. Tda1 phosphorylated by Snf1 directly phosphorylates Hxk2.**

(A) The indicated yeast strains were grown to log phase in YPD medium. Subsequently, the medium was switched from YPD to YP with 0.05% glucose, and cells were harvested at the indicated time points. Total cell lysates were resolved by Phos-tag SDS-PAGE (to detect Hxk2) or SDS-PAGE (to detect Tda1 and Cdc48) and analyzed via immunoblotting using anti-Hxk2, anti-Tda1, and anti-Cdc48 antibodies to ensure uniform loading. (B) The relative levels of phosphorylated Hxk2 to total Hxk2 were quantified. Error bars represent mean ± standard deviation (SD) of three independent experiments. $P$ value was calculated by one-way analysis of variance (ANOVA) with Dunnett's test. The $P$ values at 10 min were as follows: WT cells versus Hxk2-S15A mutant, $P < 0.0001$; WT cells versus $tda1\Delta$ mutant, $P < 0.0001$; WT cells versus $snf1\Delta$ mutant, $P = 0.0005$. At 20 min, the $P$ values were as follows: WT cells versus Hxk2-S15A, $P < 0.0001$; WT cells versus $tda1\Delta$ mutant, $P < 0.0001$; WT cells versus $snf1\Delta$ mutant, $P < 0.0001$. (C) The Tda1 bands were divided into three categories based on their mobility: Low (L), Middle (M) and High (H). The relative levels of low mobility Tda1 (L) to total Tda1 were quantified. Error bars represent mean ± SD of three independent experiments. $P$ value was calculated by one-way analysis of variance (ANOVA) with Dunnett's test. The $P$ value at 10 min was as follows: WT cells versus $snf1\Delta$ mutant, $P = 0.0043$. At 20 min, the $P$ value was as follows: WT cells versus $snf1\Delta$ mutant, $P < 0.0001$. (D) WT cells were grown to log phase in YPD medium, followed by a 30-minute incubation in YP medium with 0.05% glucose. Total cell lysates were treated alone, with λ protein phosphatase, or with λ protein phosphatase and a phosphatase inhibitor, then subjected to immunoblotting with anti-Tda1 and anti-Cdc48 antibodies. (E) The indicated yeast strains, grown to log phase in YPD medium or in YP medium with 0.05% glucose for another 30 min were harvested and lysed using glass beads and a Micro Smash MS-100R. Lysates were subjected to immunoprecipitation (IP) with anti-HA antibody, and the resulting precipitates, along with total cell lysates, were analyzed via immunoblotting with anti-HA and anti-FLAG antibodies. (F–I) Lysates from the indicated strains were prepared as described in (E) and immunoprecipitated using anti-HA antibody. Beads bound to C-terminally 3xHA-tagged Snf1 (F, G), Snf1-K84R (F), Tda1 (G, H) or Tda1-K68R (H) were incubated with recombinant Tda1-FLAG-His$_6$ (F), Hxk2-Myc-His$_6$ (G–I), or Hxk2-S15A-Myc-His$_6$ (I) in the presence of ATP and MgCl$_2$. After the reaction, the samples were resolved by Phos-tag SDS-PAGE (to detect Tda1-FLAG-His$_6$ and Hxk2-Myc-His$_6$) or SDS-PAGE (to detect Tda1-3xHA and Snf1-3xHA) and analyzed via immunoblotting using anti-FLAG (F), anti-Myc (G–I), and anti-HA (F–I) antibodies. Source data are available online for this figure.

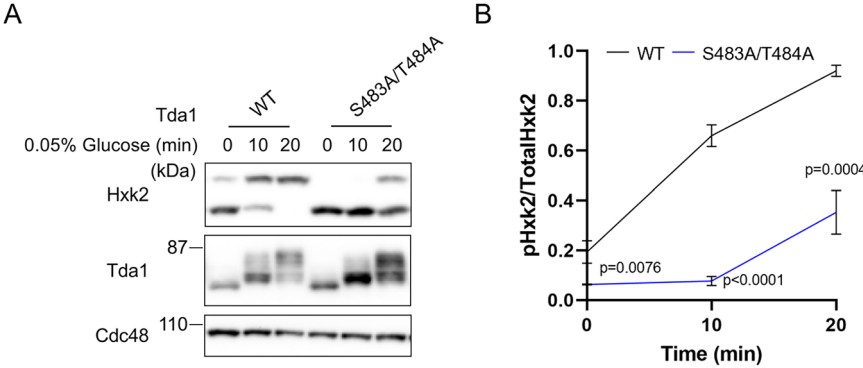

**Figure 2. The S483 and T484 residues of Tda1 are crucial for Hxk2 phosphorylation.**

(A) Lysates from the indicated strains were prepared and analyzed via immunoblotting as described in Fig. 1A. (B) The relative levels of phosphorylated Hxk2 to total Hxk2 were quantified. Error bars represent mean ± SD of three independent experiments. $P$ value was calculated by unpaired two-tailed $t$ test. The $P$ value at 0 min was as follows: WT cells versus Tda1-S483A/T484A mutant, $P = 0.0076$. At 10 min, the $P$ value was as follows: WT cells versus Tda1-S483A/T484A mutant, $P < 0.0001$. At 20 min, the $P$ value was as follows: WT cells versus Tda1-S483A/T484A mutant, $P = 0.0004$. Source data are available online for this figure.

confirming that the Ser483 and Thr484 residues are crucial for Tda1's role in Hxk2 phosphorylation.

## Yeast GSK3-β is involved in the phosphorylation of Hxk2 through Tda1

Tda1 isolated from *snf1Δ* mutants exhibited reduced in vitro Hxk2 phosphorylation activity compared to Tda1 from wild-type (WT) cells, although it retained some activity (Fig. 1G). This suggests that in addition to Snf1, another unidentified factor may regulate Tda1's Hxk2 phosphorylation activity. To explore this possibility, we immunoprecipitated Tda1 from cells cultured in low-glucose medium and analyzed Tda1-associated proteins using mass spectrometry. Among the 25 identified kinases (Appendix Table S1), we focused on the GSK3-β homologs: Rim11, Mck1, and Mrk1. Notably, previous comprehensive analyses have suggested that Rim11 interacts with Tda1 (Fasolo et al, 2011). This interaction was confirmed through co-immunoprecipitation experiments under both high- and low-glucose conditions (Fig. 3A).

Yeast contains four homologs of mammalian GSK3-β: Rim11, Mrk1, Mck1, and Ygk3. It has been reported that under glucose starvation conditions, the double knockout mutant of *RIM11* and *MCK1* and the quadruple knockout mutant of *GSK3-β* display similar phenotypes (Hirata et al, 2003). Additionally, comprehensive analyses of intracellular protein abundance have revealed that the levels of Mrk1 and Ygk3 are significantly lower compared to those of Rim11 and Mck1 (Ho et al, 2018). Based on these findings, we considered Rim11 and Mck1 as the primary GSK3-β proteins and investigated their role in regulating Tda1 activity. Under low-glucose conditions, the levels of phosphorylated Tda1 were slightly reduced in the *rim11Δ* and *mck1Δ* mutants compared to WT cells, and significantly decreased in the *rim11Δmck1Δ* and kinase dead Rim11-K68R *mck1Δ* mutants (Fig. 3B,D; Appendix Fig. S2E,F). Similarly, Hxk2 phosphorylation was also slightly reduced in the *rim11Δ* and *mck1Δ* single mutants and significantly reduced in the *rim11Δmck1Δ* mutant (Fig. 3B,C).

Next, we immunoprecipitated Rim11 from yeast cells grown in low-glucose medium and evaluated its ability to phosphorylate Tda1 in vitro (Fig. 3E). Our results indicate that Rim11

phosphorylated Tda1. However, Rim11-K68R mutant failed to phosphorylate Tda1. We then assessed whether yeast GSK3-β activity is required for the in vitro phosphorylation of Hxk2 by Tda1 (Fig. 3F). Tda1 from the *rim11Δmck1Δ* mutant showed a significant reduction in Hxk2 phosphorylation compared to Tda1 from WT cells. Additionally, we explored whether yeast GSK3-β directly phosphorylates Hxk2. In phosphorylation assays using Rim11 purified from low-glucose medium, no Hxk2 phosphorylation was detected, regardless of the presence of Tda1. These findings imply that Tda1, when phosphorylated and activated by yeast GSK3-β, is responsible for phosphorylating Hxk2.

We then investigated which amino acid residues of Tda1 are involved in Hxk2 phosphorylation by yeast GSK3-β. Tda1 was immunoprecipitated from cells grown in YPD medium or YP medium with 0.05% glucose, and the phosphorylated residues were analyzed using mass spectrometry (Dataset EV1). Peptides corresponding to residues 393-418 and 494-515 of Tda1 were identified, each containing phosphorylated Ser393 and Thr397, as well as Thr504, Ser507 or Ser509, and Thr513. The Ser393 and Ser509 residues corresponded to the consensus phosphorylation motifs, specifically the S/TxxxpT sequence recognized by yeast GSK3-β (Mok et al, 2010). To determine if the Ser393 and Ser509 residues of Tda1 are critical for Hxk2 phosphorylation, we examined Hxk2 phosphorylation levels in the Tda1-S393A and S509A mutants under low-glucose conditions (Fig. 3G,H). The Tda1-S509A mutant showed reduced levels of phosphorylated Hxk2 compared to WT cells, while the Tda1-S393A mutant did not exhibit a similar decrease. Additionally, the Tda1-T513A mutant displayed reduced Hxk2 phosphorylation compared to WT cells (Fig. 3J,K). Correspondingly, Tda1 phosphorylation was not reduced in the Tda1-S393A mutant but was significantly diminished in the Tda1-S509A and T513A mutants compared to WT cells (Fig. 3G,I,J,L). Given that yeast GSK3-β requires the Thr residue located four residues downstream of the target to be pre-phosphorylated, these results strongly suggest that Ser509 of Tda1 is a target of yeast GSK3-β. Therefore, these findings indicate that yeast GSK3-β regulates Hxk2 phosphorylation through the phosphorylation of the Ser509 residue of Tda1.

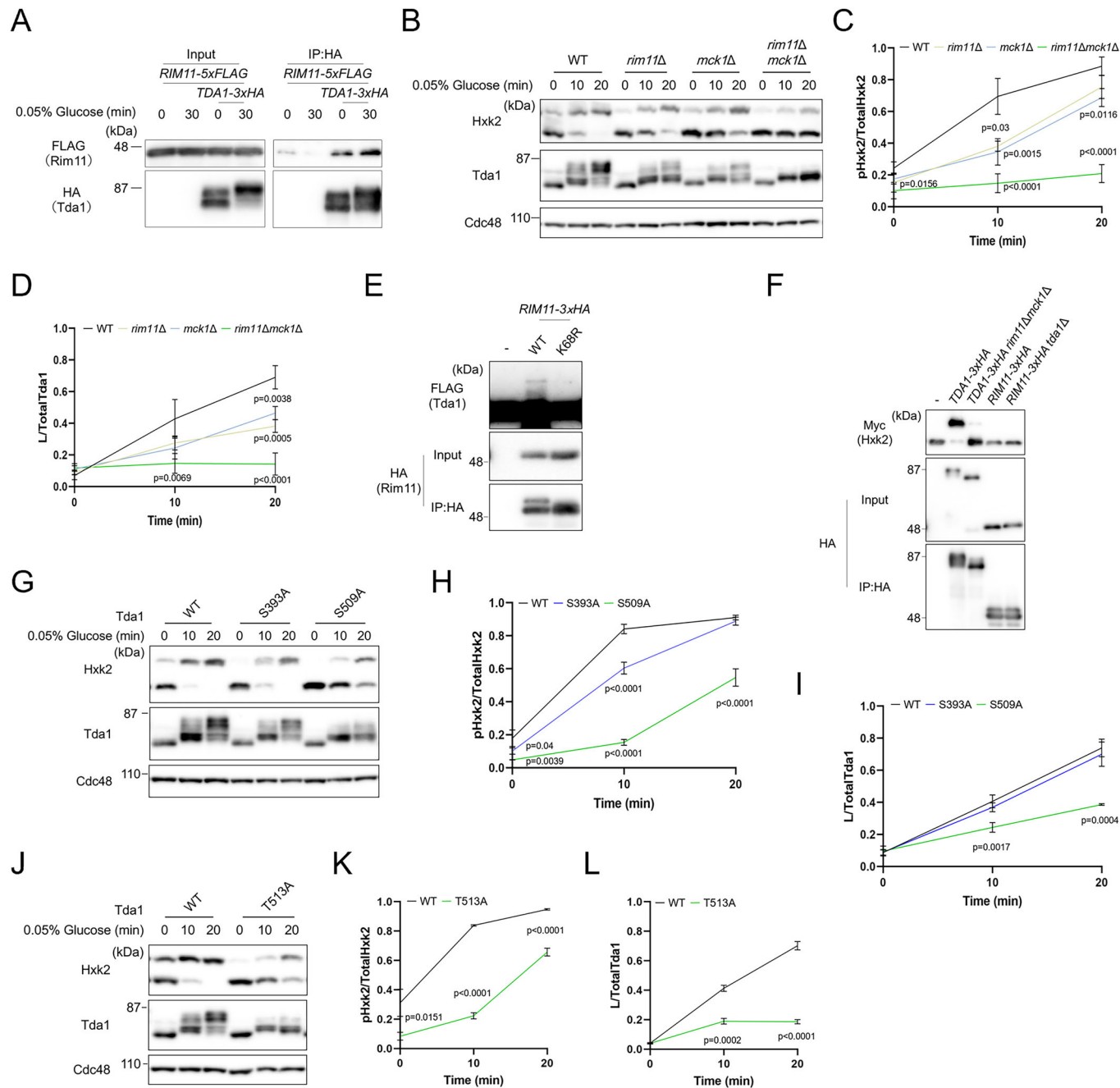

Next, we aimed to identify the kinases responsible for phosphorylating Tda1 at the Thr513 residue. Our mass spectrometry analyses revealed that Tda1 can bind to four MAPKs: Kss1, Fus3, Slt2, and Hog1. MAPKs typically phosphorylate Ser and Thr residues located immediately before a Pro residue (Mok et al, 2010). Given that Tda1 contains the Pro514 residue, we examined the effects of deletion mutants of these kinases on Hxk2 phosphorylation. To account for potential functional redundancy, we also analyzed simultaneous deletion mutants of these kinases (Appendix Fig. S3A–E). However, we did not observe any mutants in which Hxk2 phosphorylation was reduced compared to WT cells. Previous studies indicate that kinases such as Pho85 and Cdc28

can phosphorylate Ser and Thr residues situated just before a Pro residue. Furthermore, Rim15 is known to phosphorylate Ser and Thr residues located three positions upstream of a Thr residue (Mok et al, 2010). Considering that Tda1 contains the Thr516 residue, we examined this as well. In the deletion mutants of *PHO85* and *RIM15*, Hxk2 phosphorylation did not decrease compared to WT cells (Appendix Fig. S2C). Since *CDC28* is an essential gene, we utilized an analog-sensitive mutant for these assays (Appendix Fig. S3F). Treatment with the ATP analog 1NM-PP1 led to an increase in Sic1 expression, which is normally phosphorylated by Cdc28 and subsequently degraded via the ubiquitin-proteasome pathway, indicating the successful

**Figure 3.    Yeast GSK3-β plays a role in Hxk2 phosphorylation through Tda1.**

(A) Lysates from the indicated strains were prepared as described in Fig. 1E and subjected to immunoprecipitation (IP) with anti-HA. The immunoprecipitates and total cell lysates were analyzed by immunoblotting using anti-HA and anti-FLAG. (B) Lysates from the indicated strains were prepared and analyzed via immunoblotting as described in Fig. 1A. (C) The relative levels of phosphorylated Hxk2 to total Hxk2 were quantified. Error bars represent mean ± SD of three independent experiments. $P$ value was calculated by one-way analysis of variance (ANOVA) with Dunnett's test. The $P$ value at 0 min was as follows: WT cells versus $rim11\Delta mck1\Delta$ mutant, $P = 0.0156$. At 10 min, the $P$ values were as follows: WT cells versus $rim11\Delta$, $P = 0.03$; WT cells versus $mck1\Delta$ mutant, $P = 0.0015$; WT cells versus $rim11\Delta mck1\Delta$ mutant, $P < 0.0001$. At 20 min, the $P$ values were as follows: WT cells versus $mck1\Delta$ mutant, $P = 0.0116$; WT cells versus $rim11\Delta mck1\Delta$ mutant, $P < 0.0001$. (D) The relative levels of low mobility Tda1 to total Tda1 were quantified. Error bars represent mean ± SD of three independent experiments. $P$ value was calculated by one-way analysis of variance (ANOVA) with Dunnett's test. The $P$ value at 10 min was as follows: WT cells versus $rim11\Delta mck1\Delta$ mutant, $P = 0.0069$. At 20 min, the $P$ values were as follows: WT cells versus $rim11\Delta$ mutant, $P = 0.0005$; WT cells versus $mck1\Delta$ mutant, $P = 0.0038$; WT cells versus $rim11\Delta mck1\Delta$ mutant, $P < 0.0001$. (E, F) Beads bound to C-terminally 3xHA-tagged Rim11 (E, F), Rim11-K68R (E) or Tda1 (F) were prepared as described in Fig. 1F and incubated with recombinant Tda1-FLAG-His$_6$ (E) or Hxk2-Myc-His$_6$ (F) in the presence of ATP and MgCl$_2$. After the reaction, samples were separated by Phos-tag SDS-PAGE (to detect Tda1-FLAG-His$_6$ and Hxk2-Myc-His$_6$) or SDS-PAGE (to detect Tda1-3xHA and Rim11-3xHA) and analyzed by immunoblotting with anti-FLAG (E), anti-Myc (F), and anti-HA (E, F). (G) Lysates from the indicated strains were prepared and analyzed via immunoblotting as described in Fig. 1A. (H) The relative levels of phosphorylated Hxk2 to total Hxk2 were quantified. Error bars represent mean ± SD of three independent experiments. $P$ value was calculated by one-way analysis of variance (ANOVA) with Dunnett's test. The $P$ values at 0 min were as follows: WT cells versus Tda1-S393A mutant, $P = 0.04$; WT cells versus Tda1-S509A mutant, $P = 0.0039$. At 10 min, the $P$ values were as follows: WT cells versus Tda1-S393A mutant, $P < 0.0001$; WT cells versus Tda1-S509A mutant, $P < 0.0001$. At 20 min, the $P$ value was as follows: WT cells versus Tda1-S509A mutant, $P < 0.0001$. (I) The relative levels of low mobility Tda1 to total Tda1 were quantified. Error bars represent mean ± SD of three independent experiments. $P$ value was calculated by one-way analysis of variance (ANOVA) with Dunnett's test. The $P$ value at 10 min was as follows: WT cells versus Tda1-S509A mutant, $P = 0.0017$. At 20 min, the $P$ value was as follows: WT cells versus Tda1-S509A mutant, $P = 0.0004$. (J) Lysates from the indicated strains were prepared and analyzed via immunoblotting as described in Fig. 1A. (K) The relative levels of phosphorylated Hxk2 to total Hxk2 were quantified. Error bars represent mean ± SD of three independent experiments. $P$ value was calculated by unpaired two-tailed $t$ test. The $P$ value at 0 min was as follows: WT cells versus Tda1-T513A mutant, $P = 0.0151$. At 10 min, the $P$ value was as follows: WT cells versus Tda1- T513A mutant, $P < 0.0001$. At 20 min, the $P$ value was as follows: WT cells versus Tda1- T513A mutant, $P < 0.0001$. (L) The relative levels of low mobility Tda1 to total Tda1 were quantified. Error bars represent mean ± SD of three independent experiments. $P$ value was calculated by unpaired two-tailed $t$ test. The $P$ value at 10 min was as follows: WT cells versus Tda1-T513A mutant, $P = 0.0002$. At 20 min, the $P$ value was as follows: WT cells versus Tda1- T513A mutant, $P < 0.0001$. Source data are available online for this figure.

inactivation of Cdc28. However, even under these conditions, Hxk2 phosphorylation remained unchanged compared to WT cells. Further research is needed to identify the kinases responsible for the phosphorylation of Tda1 at the Thr513 residue.

### Snf1 and yeast GSK3-β independently contribute to the Tda1-mediated phosphorylation of Hxk2 at Ser15

We further investigated the mechanisms by which Snf1 and yeast GSK3-β regulate Hxk2 phosphorylation. Under low-glucose conditions, the phosphorylation level of Hxk2 in the $snf1\Delta rim11\Delta mck1\Delta$ mutant was lower than that observed in the $snf1\Delta$ and $rim11\Delta mck1\Delta$ mutants (Fig. 4A,B). This finding suggests that Snf1 and yeast GSK3-β enhance Hxk2 phosphorylation through distinct pathways.

Additionally, we evaluated the necessity of yeast GSK3-β and Snf1 for the in vitro phosphorylation of Hxk2 at Ser15 by Tda1 (Fig. 4C). Tda1 from $snf1\Delta$ and $rim11\Delta mck1\Delta$ mutants exhibited moderate and significant reductions in Hxk2 phosphorylation activity at Ser15, respectively, compared to Tda1 from WT cells. Moreover, no Hxk2 phosphorylation activity was detected in Tda1 isolated from $snf1\Delta rim11\Delta mck1\Delta$ mutants. These results underscore that Snf1 and yeast GSK3-β act independently in facilitating Tda1-mediated phosphorylation of Hxk2 at Ser15.

### The Ser483 and Thr484 residues and the Ser509 or Thr513 residues of Tda1 additively regulate the phosphorylation of Hxk2 at Ser15

Next, we examined the phosphorylation levels of Hxk2 at Ser15 in a mutant where Ser483, Thr484, and Ser509 were all substituted with alanine (Fig. 4D,E). As anticipated, under low-glucose conditions, the phosphorylation level of Hxk2 was significantly lower in the Tda1-S483A/T484A/S509A mutant compared to the Tda1-S483A/

T484A and Tda1-S509A mutants. A similar reduction was noted in the Tda1-S483A/T484A/T513A mutant (Fig. 4F,G). These findings indicate that the phosphorylation of Hxk2 at Ser15 is regulated additively by the Ser483 and Thr484 residues, as well as by the Ser509 or Thr513 residues of Tda1.

### The Ser483 and Thr484 residues of Tda1 are crucial for its activation by Snf1, while the Ser509 residue is vital for its activation by yeast GSK3-β

We then explored whether Snf1 and yeast GSK3-β regulate Hxk2 phosphorylation via the Ser483/Thr484 and Ser509 residues of Tda1, respectively. The Tda1-S509A $snf1\Delta$ mutant showed reduced Hxk2 phosphorylation compared to both the Tda1-S509A and $snf1\Delta$ mutants (Fig. 5A,B). Similarly, the Tda1-S483A/T484A $rim11\Delta mck1\Delta$ mutant exhibited lower levels of Hxk2 phosphorylation than the Tda1-S483A/T484A and $rim11\Delta mck1\Delta$ mutants (Fig. 5C,D). These results indicate that the effects of the S483A/T484A and S509A mutations in Tda1 on Hxk2 phosphorylation parallel the effects of inactivating Snf1 and yeast GSK3-β, respectively.

### Tda1 functions as a negative regulator of the glucose starvation signaling pathway

Next, we investigated whether Tda1 affects cell proliferation when glycerol and acetate, non-fermentable carbon sources, are used instead of glucose (Fig. 6A). The $snf1\Delta$ mutant displayed growth defects on plates containing glycerol and acetate. Notably, the $tda1\Delta$ and Hxk2-S15A mutants grew more rapidly than WT cells on glycerol and acetate plates. To further investigate these findings, growth rates were measured in YPD medium and YP medium with 3% glycerol (Fig. 6B). Consistent with the growth patterns on plates (Fig. 6A), the $tda1\Delta$ and Hxk2-S15A mutants exhibit enhanced

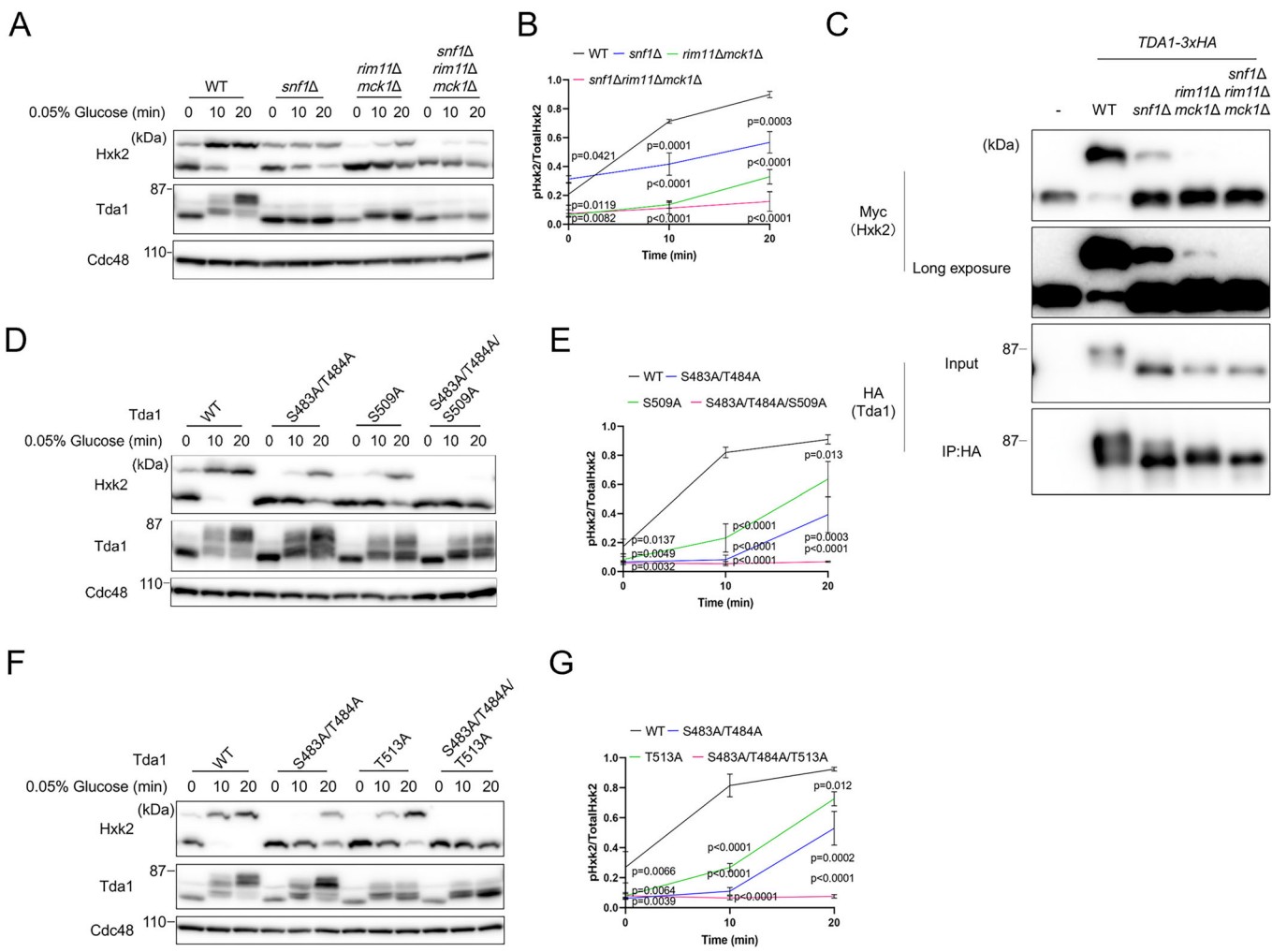

**Figure 4. Snf1 and yeast GSK3-β are independently involved in the phosphorylation of Hxk2 via Tda1.**

(A) Lysates from the indicated strains were prepared and subjected to immunoblotting as described in Fig. 1A. (B) The relative levels of phosphorylated Hxk2 to total Hxk2 were quantified. Error bars represent mean ± SD of three independent experiments. P value was calculated by one-way analysis of variance (ANOVA) with Dunnett's test. The P values at 0 min were as follows: WT cells versus snf1Δ mutant, P = 0.0421; WT cells versus rim11Δmck1Δ mutant, P = 0.0082; WT cells versus snf1Δrim11Δmck1Δ mutant, P = 0.0119. At 10 min, the P values were as follows: WT cells versus snf1Δ mutant, P = 0.0001; WT cells versus rim11Δmck1Δ mutant, P < 0.0001; WT cells versus snf1Δrim11Δmck1Δ mutant, P < 0.0001. At 20 min, the P values were as follows: WT cells versus snf1Δ mutant, P = 0.0003; WT cells versus rim11Δmck1Δ mutant, P < 0.0001; WT cells versus snf1Δrim11Δmck1Δ mutant, P < 0.0001. (C) Beads bound to C-terminally 3xHA-tagged Tda1 were prepared as described in Fig. 1F and incubated with recombinant Hxk2-Myc-His$_6$ in the presence of ATP and MgCl$_2$. After the reaction, the samples were resolved by Phos-tag SDS-PAGE (for detecting Hxk2-Myc-His$_6$) or SDS-PAGE (for detecting Tda1-3xHA) and analyzed by immunoblotting with anti-Myc and anti-HA antibodies. (D) Lysates from the indicated strains were prepared and subjected to immunoblotting as described in Fig. 1A. (E) The relative levels of phosphorylated Hxk2 to total Hxk2 were quantified. Error bars represent mean ± SD of three independent experiments. P value was calculated by one-way analysis of variance (ANOVA) with Dunnett's test. The P values at 0 min were as follows: WT cells versus Tda1-S483A/T484A mutant, P = 0.0049; WT cells versus Tda1-S509A mutant, P = 0.0137; WT cells versus Tda1- S483A/T484A/S509A mutant, P = 0.0032. At 10 min, the P values were as follows: WT cells versus Tda1-S483A/T484A mutant, P < 0.0001; WT cells versus Tda1-S509A mutant, P < 0.0001; WT cells versus Tda1- S483A/T484A/S509A mutant, P < 0.0001. At 20 min, the P values were as follows: WT cells versus Tda1-S483A/T484A mutant, P = 0.0003; WT cells versus Tda1-S509A mutant, P = 0.013; WT cells versus Tda1- S483A/T484A/S509A mutant, P < 0.0001. (F) Lysates from the indicated strains were prepared and subjected to immunoblotting as described in Fig. 1A. (G) The relative levels of phosphorylated Hxk2 to total Hxk2 were quantified. Error bars represent mean ± SD of three independent experiments. P value was calculated by one-way analysis of variance (ANOVA) with Dunnett's test. The P values at 0 min were as follows: WT cells versus Tda1-S483A/T484A mutant, P = 0.0039; WT cells versus Tda1-T513A mutant, P = 0.0066; WT cells versus Tda1- S483A/T484A/T513A mutant, P = 0.0064. At 10 min, the P values were as follows: WT cells versus Tda1-S483A/T484A mutant, P < 0.0001; WT cells versus Tda1-T513A mutant, P < 0.0001; WT cells versus Tda1- S483A/T484A/T513A mutant, P < 0.0001. At 20 min, the P values were as follows: WT cells versus Tda1-S483A/T484A mutant, P = 0.0002; WT cells versus Tda1-T513A mutant, P = 0.012; WT cells versus Tda1- S483A/T484A/T513A mutant, P < 0.0001. Source data are available online for this figure.

growth compared to WT, while the snf1Δ and rim11Δmck1Δ mutants show suppressed growth. Next, to examine whether the absence of Hxk2 phosphorylation rescues the growth defect in the snf1Δ mutant or, conversely, whether Snf1 activity is necessary for

the enhanced growth observed in tda1Δ or Hxk2-S15A mutants, we generated snf1Δtda1Δ and snf1Δ Hxk2-S15A mutants and conducted spot assays (Appendix Fig. S4). The snf1Δtda1Δ and snf1Δ Hxk2-S15A mutants exhibited significantly slower growth

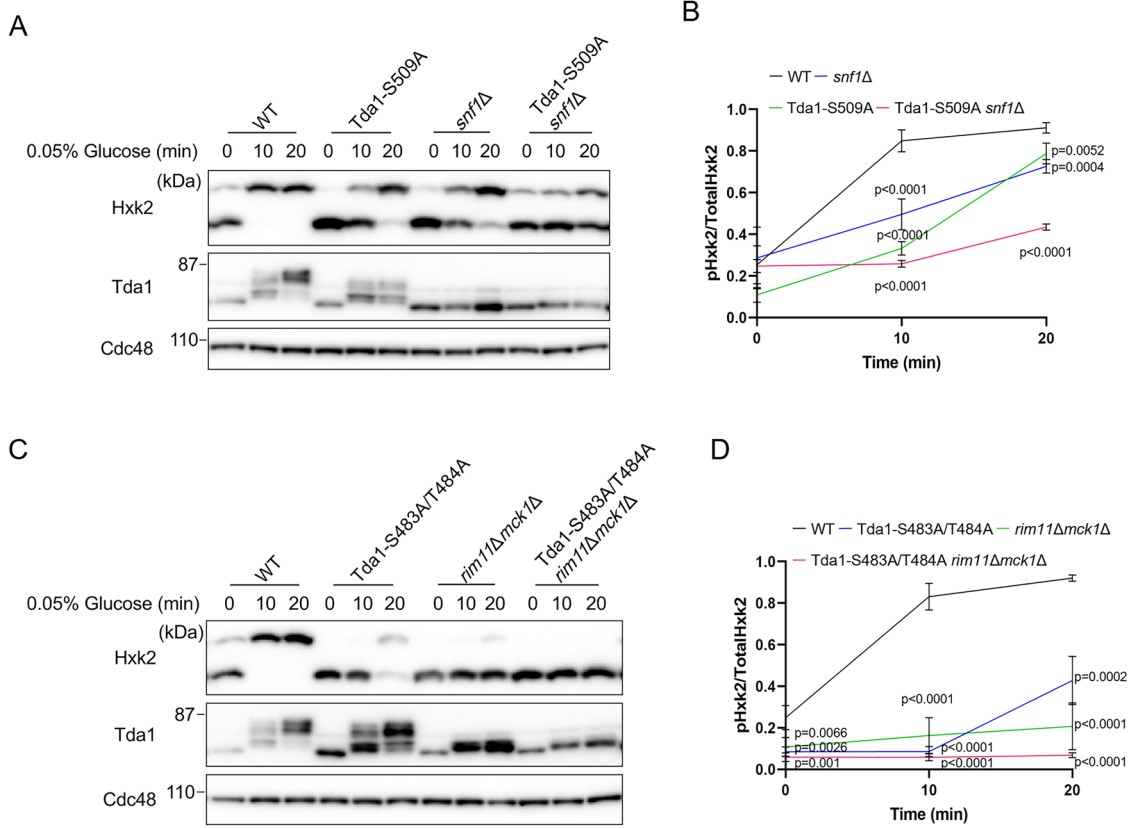

**Figure 5. The Ser483 and Thr484 residues of Tda1 are important for its activation by Snf1, while the Ser509 residue is crucial for its activation by yeast GSK3-β.**

(A) Lysates from the indicated strains were prepared and analyzed via immunoblotting as described in Fig. 1A. (B) The relative levels of phosphorylated Hxk2 to total Hxk2 were quantified. Error bars represent mean ± SD of three independent experiments. *P* value was calculated by one-way analysis of variance (ANOVA) with Dunnett's test. The *P* values at 10 min were as follows: WT cells versus *snf1Δ* mutant, *P* < 0.0001; WT cells versus Tda1-S509A mutant, *P* < 0.0001; WT cells versus Tda1- S509A *snf1Δ* mutant, *P* < 0.0001. At 20 min, the *P* values were as follows: WT cells versus *snf1Δ* mutant, *P* = 0.0004; WT cells versus Tda1-S509A mutant, *P* = 0.0052; WT cells versus Tda1- S509A *snf1Δ* mutant, *P* < 0.0001. (C) Lysates from the indicated strains were prepared and analyzed via immunoblotting as described in Fig. 1A. (D) The relative levels of phosphorylated Hxk2 to total Hxk2 were quantified. Error bars represent mean ± SD of three independent experiments. *P* value was calculated by one-way analysis of variance (ANOVA) with Dunnett's test. The *P* values at 0 min were as follows: WT cells versus Tda1-S483A/T484A mutant, *P* = 0.0026; WT cells versus *rim11Δmck1Δ* mutant, *P* = 0.0066; WT cells versus Tda1- S483A/T484A *rim11Δmck1Δ* mutant, *P* = 0.001. At 10 min, the *P* values were as follows: WT cells versus Tda1-S483A/T484A mutant, *P* < 0.0001; WT cells versus *rim11Δmck1Δ* mutant, *P* < 0.0001; WT cells versus Tda1- S483A/T484A *rim11Δmck1Δ* mutant, *P* < 0.0001. At 20 min, the *P* values were as follows: WT cells versus Tda1-S483A/T484A mutant, *P* = 0.0002; WT cells versus *rim11Δmck1Δ* mutant, *P* < 0.0001; WT cells versus Tda1- S483A/ T484A *rim11Δmck1Δ* mutant, *P* < 0.0001. Source data are available online for this figure.

compared to the *tda1Δ* and Hxk2-S15A mutants on glycerol plates, indicating that Snf1 activity is required for the enhanced growth observed in the *tda1Δ* and Hxk2-S15A mutants.

To explore this further, we cultured WT, *snf1Δ*, *tda1Δ*, and Hxk2-S15A cells in YP medium with 3% glycerol and performed transcriptome analyses using RNA-seq. We first compared gene expression levels using heat maps and scatter plots (Fig. 6C,D). Heat maps were created using the top 600 genes with the highest expression variation. Surprisingly, we observed a negative correlation between the *snf1Δ* and *tda1Δ* mutants, as well as between the *snf1Δ* and Hxk2-S15A mutants. In contrast, there was a positive correlation between the *tda1Δ* and Hxk2-S15A mutants. These points were further clarified by principal component analysis (Appendix Fig. S5). Next, we examined gene expressions across various metabolic pathways (Fig. 6E; Dataset EV2) (Oh et al, 2018). In gluconeogenesis, the TCA cycle, and oxidative phosphorylation, the expression levels of many genes in the *tda1Δ* and Hxk2-S15A

mutants were higher than those in the WT cells. In contrast, the expression levels in the *snf1Δ* mutant were lower compared to WT cells. This trend is consistent with the growth patterns of cells shown in Fig. 6A,B. Furthermore, in gluconeogenesis, the TCA cycle, and oxidative phosphorylation, we again observed a negative correlation between the *snf1Δ* and *tda1Δ* mutants, as well as between the *snf1Δ* and Hxk2-S15A mutants. Conversely, a strong positive correlation was noted between the *tda1Δ* and Hxk2-S15A mutants, although these correlations were less pronounced in glucose fermentation.

We then selected three glucose repression genes—*PCK1*, *FBP1*, and *CIT3*—and analyzed their expression levels using real-time qPCR (Fig. 6F). Consistent with the RNA-seq results, the *tda1Δ* and Hxk2-S15A mutants exhibited opposing regulation of these genes compared to the *snf1Δ* mutant, relative to WT cells. Yeast GSK3-β is known to control the stress-responsive transcription factors Msn2 and Msn4, which play a role in the glucose repressed genes

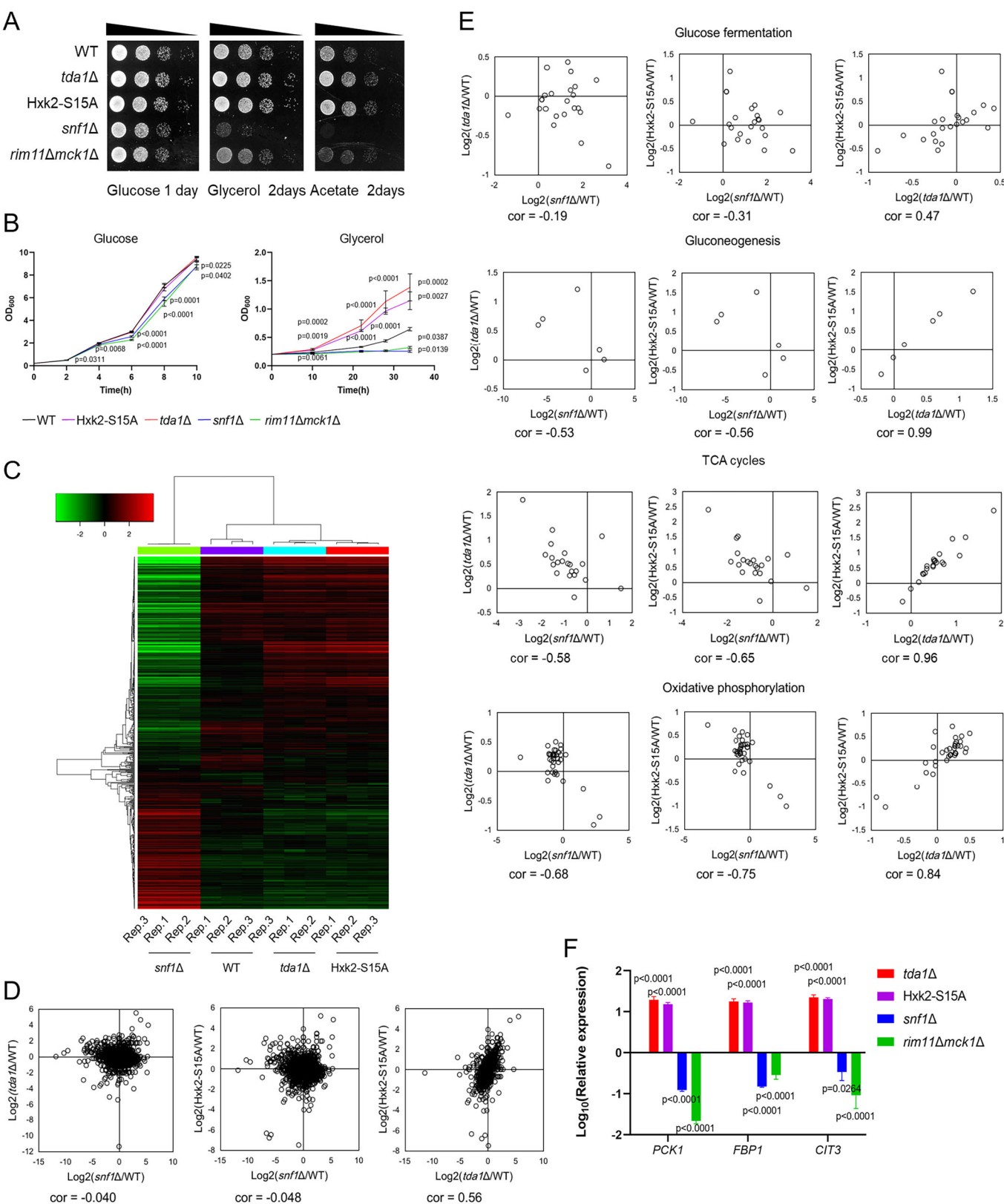

◄ **Figure 6.  Tda1 functions as a negative regulator of the glucose starvation signaling pathway.**

(A) The indicated strains were grown to log phase, adjusted to a cell density of $OD_{600} = 0.3$, serially diluted 1:10, spotted onto plates containing the specified carbon sources, and incubated at 30 °C for the indicated durations. (B) The indicated strains were grown to log phase and adjusted to a cell density of $OD_{600} = 0.2$. The cells were then transferred to YP medium containing either 2% glucose or 3% glycerol, and the $OD_{600}$ was measured at the indicated time points. Error bars represent mean ± SD of three independent experiments. *P* value was calculated by one-way analysis of variance (ANOVA) with Dunnett's test. The *P* value at 2 h in glucose medium was as follows: WT cells versus *rim11Δmck1Δ* mutant, *P* = 0.0311. At 4 h in glucose medium, the *P* value was as follows: WT cells versus *rim11Δmck1Δ* mutant, *P* = 0.0068. At 6 h in glucose medium, the *P* values were as follows: WT cells versus *snf1Δ* mutant, *P* < 0.0001; WT cells versus *rim11Δmck1Δ* mutant, *P* < 0.0001. At 8 h in glucose medium, the *P* values were as follows: WT cells versus *snf1Δ* mutant, *P* = 0.0001; WT cells versus *rim11Δmck1Δ* mutant, *P* < 0.0001. At 10 h in glucose medium, the *P* values were as follows: WT cells versus *snf1Δ* mutant, *P* = 0.0225; WT cells versus *rim11Δmck1Δ* mutant, *P* = 0.0402. The *P* values at 10 h in glycerol medium were as follows: WT cells versus Hxk2-S15A mutant, *P* = 0.0019; WT cells versus *tda1Δ* mutant, *P* = 0.0002; WT cells versus *rim11Δmck1Δ* mutant, *P* = 0.0061. At 22 h in glycerol medium, the *P* values were as follows: WT cells versus Hxk2-S15A mutant, *P* < 0.0001; WT cells versus *tda1Δ* mutant, *P* < 0.0001. At 28 h in glycerol medium, the *P* values were as follows: WT cells versus Hxk2-S15A mutant, *P* = 0.0001; WT cells versus *tda1Δ* mutant, *P* < 0.0001. At 34 h in glycerol medium, the *P* values were as follows: WT cells versus Hxk2-S15A mutant, *P* = 0.0027; WT cells versus *tda1Δ* mutant, *P* = 0.0002; WT cells versus *snf1Δ* mutant, *P* = 0.0139; WT cells versus *rim11Δmck1Δ* mutant, *P* = 0.0387. (C) After culturing the indicated strains for 6 h in YP medium with 3% glycerol, the differentially expressed genes were analyzed by RNA-seq. For the top 600 genes with the highest variance, CPM normalized count values were log2 transformed and clustered using the average linkage method with Pearson correlation. Heatmap was visualized using iDEP v.0.96. The color code indicates the difference from the average of the transformed counts across three independent experiments. (D) RNA-seq scatter plots showing transcripts of all genes. cor indicates correlation coefficient. (E) RNA-seq scatter plots showing transcripts of genes involved in the indicated pathways. (F) The indicated strains were grown to log phase in YPD medium and further incubated for 6 h in YP medium with 3% glycerol. Total RNA was extracted and analyzed by RT-qPCR. The relative expression levels of the *PCK1*, *FBP1*, and *CIT3* genes were normalized to the *ACT1* gene. Error bars represent mean ± SD of three independent experiments. *P* value was calculated by one-way analysis of variance (ANOVA) with Dunnett's test. The *P* values for *PCK1* relative expression levels are as follows: WT cells versus Hxk2-S15A mutant, *P* < 0.0001; WT cells versus *tda1Δ* mutant, *P* < 0.0001; WT cells versus *snf1Δ* mutant, *P* < 0.0001; WT cells versus *rim11Δmck1Δ* mutant, *P* < 0.0001. For *FBP1*, the *P* values were as follows: WT cells versus Hxk2-S15A mutant, *P* < 0.0001; WT cells versus *tda1Δ* mutant, *P* < 0.0001; WT cells versus *snf1Δ* mutant, *P* < 0.0001; WT cells versus *rim11Δmck1Δ* mutant, *P* < 0.0001. For *CIT3*, the *P* values were as follows: WT cells versus Hxk2-S15A mutant, *P* < 0.0001; WT cells versus *tda1Δ* mutant, *P* < 0.0001; WT cells versus *snf1Δ* mutant, *P* = 0.0264; WT cells versus *rim11Δmck1Δ* mutant, *P* < 0.0001. Source data are available online for this figure.

(Hirata et al, 2003). Given that the target genes of Msn2 and Msn4 partially overlap with those downstream of Snf1 (Kuang et al, 2017), we also analyzed gene expression levels in *rim11Δmck1Δ* mutants (Fig. 6F). The gene expression patterns in the *rim11Δmck1Δ* mutant closely resembled those observed in the *snf1Δ* mutant. Moreover, we noted growth defects in the *rim11Δmck1Δ* mutant on glycerol and acetate plates (Fig. 6A). These findings suggest that Tda1 negatively regulates glucose starvation signaling mediated by Snf1 and yeast GSK-3β.

# Discussion

In this study, we sought to investigate the complex molecular mechanisms underlying the glucose starvation response in *Saccharomyces cerevisiae*, focusing on the roles of four proteins, Tda1, Hxk2, Snf1 and yeast GSK-3β in glucose metabolism. Our findings reveal intricate regulatory relationships between these proteins and suggest that Tda1, activated by both Snf1 and yeast GSK-3β, phosphorylates Hxk2 at Ser15 and functions as a negative regulator in the glucose starvation pathway.

Our study demonstrated that under low-glucose conditions, Tda1 is phosphorylated and activated by both Snf1 and yeast GSK-3β, which subsequently enables Tda1 to phosphorylate Hxk2 at Ser15. The role of Snf1 in regulating Tda1 was well established, as Snf1-mediated phosphorylation at Ser483 and Thr484 was found to be crucial for Tda1 activation (Oh et al, 2020). In contrast, the involvement of yeast GSK-3β in this process was less clear prior to this study. Our data indicated that yeast GSK-3β, specifically the homologs Rim11 and Mck1, activates Tda1 through the Ser509 residue, thereby playing a significant role in Tda1's kinase activity toward Hxk2. This dual regulation by Snf1 and yeast GSK-3β is critical for coordinating the cellular response to glucose starvation, where both pathways converge to regulate energy production and consumption.

The discovery that yeast GSK-3β is involved in the glucose starvation response adds to the growing body of evidence that suggests a close functional relationship between AMPK (or Snf1 in yeast) and GSK-3β across eukaryotes. Both kinases have long been known to play crucial roles in energy homeostasis, with AMPK acting as a cellular energy sensor and GSK-3β involved in various metabolic processes (Hardie et al, 2012; Summers et al, 1999; Xiao et al, 2007; Xiao et al, 2011). In mammals, AMPK and GSK-3β regulate overlapping substrates, including glycogen synthase, suggesting an evolutionary conserved interaction between these two kinases.

Our study demonstrated that yeast GSK-3β activates Tda1 under glucose starvation. An important question is how yeast GSK-3β gains the ability to activate Tda1 under these conditions. Previous reports have shown that Rim11 and Mck1 are inhibited by PKA, which in turn is activated by cAMP (Quan et al, 2015; Rubin-Bejerano et al, 2004). Since cAMP levels decrease under low-glucose conditions, yeast GSK-3β may become active in response to glucose starvation. Additionally, previous studies have indicated that GSK-3β's substrate phosphorylation requires prior 'priming phosphorylation' by other kinases. It is possible that as glucose levels decline, phosphorylation of Thr513 in Tda1 enables GSK-3β to phosphorylate additional sites on Tda1. However, in this study, the kinase responsible for Thr513 phosphorylation remains unidentified, and the significance of this phosphorylation event is unclear. This suggests that the full regulatory mechanism of Tda1 via yeast GSK-3β has yet to be elucidated. Future research, incorporating the priming phosphorylation layer, will likely reveal that the regulation of Tda1 is even more complex than currently understood.

One of the surprising findings of this study was that Tda1 appears to act as a negative regulator of the glucose starvation response. The deletion of *TDA1* resulted in enhanced expression of respiratory genes and improved growth under glucose-limiting

conditions, in contrast to the growth defects observed in *snf1Δ* and *rim11Δmck1Δ* mutants. Although Snf1 and yeast GSK3-β positively regulate Tda1 activity, the role of Tda1 appears paradoxical, as it suppresses rather than enhances the glucose starvation response. A possible explanation is that Snf1 promotes this response by phosphorylating multiple downstream transcriptional activators, including Adr1, Cat8, and Sip4, while yeast GSK3-β does so through factors such as Msn2 and Msn4. In contrast, Tda1, once activated by Snf1 and yeast GSK3-β, seems to act as a fine-tuning regulator, preventing excessive metabolic activation.

Our transcriptome analyses further supported this hypothesis, revealing a negative correlation between gene expression in *tda1Δ* and *snf1Δ* mutants. This indicates that Tda1 likely counteracts some of the metabolic shifts promoted by Snf1, adding an additional layer of control to the glucose starvation response. Moreover, the similarity in gene expression patterns between the *tda1Δ* and Hxk2-S15A mutants suggests that Tda1's role in repressing respiratory genes is mediated, at least in part, through its phosphorylation of Hxk2. These findings raise intriguing questions about the broader role of Tda1 in regulating cellular metabolism, particularly in balancing energy production and consumption under nutrient-limiting conditions.

Recent studies have shown that transcriptomic differences between *hxk2Δ* and wild-type strains are minor under low-glucose condition (Lesko et al, 2023). This raises the question of why the Hxk2-S15A mutant enhances gene expression. One possibility is that hexokinase activity affects the metabolic profile,

influencing transcription. Unlike *hxk2Δ* cells, which lack hexokinase activity, Hxk2-S15A retains enzymatic function but lacks Ser15 phosphorylation. This may alter glycolytic flux and metabolic balance, leading to gene expression changes. However, the precise mechanism remains unclear, requiring further study to understand how the phosphorylation of Hxk2 at Ser15 regulates transcription under glucose starvation.

Future studies could also explore the broader implications of Tda1-mediated regulation in other stress responses, as well as its potential interactions with other nutrient-sensing pathways. Given the conservation of AMPK and GSK3-β across eukaryotes, it would be interesting to investigate whether similar regulatory mechanisms exist in other organisms and how they contribute to the maintenance of energy homeostasis in different environmental contexts.

In conclusion, our findings demonstrate the complex interplay between Tda1, Hxk2, Snf1, and yeast GSK3-β in regulating the glucose starvation response in *S. cerevisiae*. Tda1 emerges as a key regulatory node, integrating signals from multiple kinases to modulate Hxk2 activity and ensure a balanced response to glucose limitation. These results contribute to our understanding of cellular energy regulation and provide a foundation for future research into the molecular mechanisms governing nutrient sensing and metabolic adaptation.

# Methods

### Reagents and tools table

| Reagent/resource | Reference or source | Identifier or catalog number |
|---|---|---|
| **Experimental models** | | |
| Yeast strains | | |
| BY4741(*MATa his3Δ1 leu2Δ0 met15Δ0 ura3Δ0*) | S. Michaelis | BY4741 |
| BY4741, *HXK2-S15A* | This study | YTK6531 |
| BY4741, *tda1Δ::KanMX4* | This study | YTK6416 |
| BY4741, *snf1Δ::KanMX4* | This study | YTK7360 |
| BY4741, *hxk2Δ::HIS3* | This study | YTK6507 |
| BY4741, *TDA1-3HA::LEU2* | This study | YTK7375 |
| BY4741, *TDA1-3HA::LEU2 SNF1-5FLAG::HIS3* | This study | YTK7985 |
| BY4741, *TDA1-K68R-3HA::LEU2* | This study | YTK8125 |
| BY4741, *TDA1-3HA::LEU2 snf1Δ::KanMX4* | This study | YTK7377 |
| BY4741, *SNF1-3HA::LEU2* | This study | YTK6425 |

| Reagent/resource | Reference or source | Identifier or catalog number |
|---|---|---|
| BY4741, *SNF1-K84R-3HA::LEU2* | This study | YTK8223 |
| BY4741, *SNF1-3HA::LEU2 tda1Δ::KanMX4* | This study | YTK6428 |
| BY4741, *SNF1-5FLAG::HIS3* | This study | YTK7984 |
| BY4741, *SNF1-5FLAG::HIS3 TDA1-3HA::LEU2* | This study | YTK7985 |
| BY4741, *TDA1-S483A/T484A* | This study | YTK6854 |
| BY4741, *RIM11-5FLAG::HIS3* | This study | YTK7582 |
| BY4741, *TDA1-3HA::LEU2 RIM11-5FLAG::HIS3* | This study | YTK7575 |
| BY4741, *rim11Δ::KanMX4* | Thermo Fisher Scientific | YTK7421 |
| BY4741, *mck1Δ::KanMX4* | Thermo Fisher Scientific | YTK7516 |
| BY4741, *rim11Δ::KanMX4 mck1Δ::URA3* | This study | YTK7504 |
| BY4741, *mck1Δ::URA3 RIM11-K68R-3HA::LEU2* | This study | YTK8194 |
| BY4741, *RIM11-3HA::LEU2* | This study | YTK7537 |
| BY4741, *RIM11-K68R-3HA::LEU2* | This study | YTK8154 |
| BY4741, *TDA1-3HA::LEU2 rim11Δ::KanMX4 mck1Δ::URA3* | This study | YTK7547 |
| BY4741, *RIM11-3HA::LEU2 tda1Δ::KanMX4* | This study | YTK7631 |
| BY4741, *TDA1- S393A* | This study | YTK7844 |
| BY4741, *TDA1- S509A* | This study | YTK7952 |
| BY4741, *TDA1- T513A* | This study | YTK7857 |
| BY4741, *snf1Δ::HPH rim11Δ::KanMX4 mck1Δ::URA3* | This study | YTK7494 |
| BY4741, *TDA1-3HA::LEU2 snf1Δ::HPH rim11Δ::KanMX4 mck1Δ::URA3* | This study | YTK7544 |
| BY4741, *TDA1-S483A/T484A/S509A* | This study | YTK7626 |
| BY4741, *TDA1-S483A/T484A/T513A* | This study | YTK7861 |
| BY4741, *TDA1-S509A snf1Δ::KanMX4* | This study | YTK7713 |

| Reagent/resource | Reference or source | Identifier or catalog number |
|---|---|---|
| BY4741, *TDA1-S483/T484A rim11Δ::KanMX4 mck1Δ::LEU2* | This study | YTK7709 |
| BY4741, *kss1Δ::KanMX4* | Thermo Fisher Scientific | YTK7742 |
| BY4741, *fus3Δ::KanMX4* | Thermo Fisher Scientific | YTK7743 |
| BY4741, *slt2Δ::KanMX4* | Thermo Fisher Scientific | YTK7744 |
| BY4741, *kss1Δ::KanMX4 fus3Δ::LEU2* | This study | YTK7848 |
| BY4741, *slt2Δ::KanMX4 fus3Δ::LEU2* | This study | YTK7850 |
| BY4741, *slt2Δ::KanMX4 kss1Δ::LEU2* | This study | YTK7852 |
| BY4741, *kss1Δ::KanMX4 fus3Δ::LEU2 slt2Δ::URA3* | This study | YTK7865 |
| BY4741, *pho85Δ::KanMX4* | Thermo Fisher Scientific | YTK4798 |
| BY4741, *rim15Δ::KanMX4* | Thermo Fisher Scientific | YTK5585 |
| BY4741, *hog1Δ::KanMX4* | Thermo Fisher Scientific | YTK5591 |
| BY4741, *kss1Δ::KanMX4 hog1Δ::HIS3* | This study | YTK7927 |
| BY4741, *fus3Δ::KanMX4 hog1Δ::HIS3* | This study | YTK7929 |
| BY4741, *slt2Δ::KanMX4 hog1Δ::HIS3* | This study | YTK7931 |
| BY4741, *kss1Δ::KanMX4 fus3Δ::LEU2 hog1Δ::HIS3* | This study | YTK7933 |
| BY4741, *slt2Δ::KanMX4 fus3Δ::LEU2 hog1Δ::HIS3* | This study | YTK7935 |
| BY4741, *slt2Δ::KanMX4 kss1Δ::LEU2 hog1Δ::HIS3* | This study | YTK7937 |
| BY4741, *kss1Δ::KanMX4 fus3Δ::LEU2 slt2Δ::URA3 hog1Δ::HIS3* | This study | YTK7939 |
| BY4741, *cdc28-as1* | This study | YTK7786 |

| Reagent/resource | Reference or source | Identifier or catalog number |
|---|---|---|
| BY4741, *tda1Δ::KanMX4 snf1Δ::LEU2* | This study | YTK6433 |
| BY4741, *HXK2-S15A snf1Δ::KanMX4* | This study | YTK8136 |
| BL21(DE3) Competent *E. coli* | New England Biolabs | C2527H |
| Mach1 Competent *E. coli* | Thermo Fisher Scientific | C862003 |
| **Recombinant DNA** | | |
| pRS306 | NovoPro Bioscience Inc. | V010232 |
| pRS306-Hxk2-S15A | This study | N/A |
| pRS306-Tda1-K68R | This study | N/A |
| pRS306-Snf1-K84R | This study | N/A |
| pRS306-Tda1-S483A/T484A | This study | N/A |
| pRS306-Rim11-K68R | This study | N/A |
| pRS306-Tda1-S393A | This study | N/A |
| pRS306-Tda1-S509A | This study | N/A |
| pRS306-Tda1-T513A | This study | N/A |
| pET23b | Sigma-Aldrich | 69746 |
| pET23b-Tda1-FLAG-His$_6$ | This study | N/A |
| pET23b-Hxk2-myc-His$_6$ | This study | N/A |
| pET23b-Hxk2-S15A-myc-His$_6$ | This study | N/A |
| **Antibodies** | | |
| Hxk2 | This study | N/A |
| Tda1 | This study | N/A |
| Cdc48 | This study | N/A |
| Sic1 | This study | N/A |
| Myc | Our laboratory | N/A |
| HA | Our laboratory | N/A |
| FLAG (M2) | Sigma-Aldrich | A8592 |
| HRP-conjugated anti-rabbit IgG | Sigma-Aldrich | A6154 |
| HA (12CA5) | Roche | 45-11583816001 |
| **Oligonucleotides and other sequence-based reagents** | | |
| Primers | Sequence (5'- > 3') | |
| HXK2-S15A-5-1033 | CAGAAAGGGTGCCATGGCCGA | N/A |
| HXK2-S15A-3-1053 | TCGGCCATGGCACCCTTTCTG | N/A |
| HXK2-5-Spe1-301 | AAGGTTACTAGTTCATGTGATTCGAACTATGAAACA | N/A |
| HXK2-3-Xho1-2850 | AGTTCCCTCGAGAGTACGCAAGCTATCTAGAGGAAG | N/A |
| HXK2-pFU-5-791 | GTGTCCAATTCCGTGATGTCTCTTTGTTGCACCTTCGCCACTGTCTTATCTCCAGATTACGCTTCTAGCT | N/A |
| HXK2-pFU-3-2511 | GTAGAAAAAGGGCACCTTCTTGTTGTTCAAACTTAATTTACAAATTAAGTCCTCGAGGCCAGAAGACTAAGAGG | N/A |
| HXK2-5-Nde1-1001 | AAGGTTCATATGGTTCATTTAGGTCCAAAAAAACCAC | N/A |

| Reagent/resource | Reference or source | Identifier or catalog number |
|---|---|---|
| HXK2-3-myc-Xho1-2458 | AGTTCCCTCGAGCAGGTCCTCCTCGGAAATCAGCTTCTGCTCAGCACCGATGATACCAACGG | N/A |
| TDA1-pFU-5-906 | CTTCAGTTACCTACCACGACTGTCTCAGCCAAACCGCGATATTTTCATACTTCATCTCCGGTTCTGCTGCTAGA | N/A |
| TDA1-pFU-5-2709 | CATTCCAGGTGAATGTTGATGATTCTGATGGTGATGAAACCATGCAGATATCCGGTTCTGCTGCTAGT | N/A |
| TDA1-pFU-3-2811 | CCCAGGTAAAAAAAAAAAGAAAGAAAAAAAAGGTGAATTATCATTCAACACCTCGAGGCCAGAAGACT | N/A |
| TDA1-5-BamHI-301 | AAGGTTGGATCCGGTTTAAGATGCTCTTTTACCAAT | N/A |
| TDA1-3-Kpn1-3150 | AGTTCCGGTACCGTAAGCTAAATGATCACTGGGCTG | N/A |
| TDA1-K68R-5-1193 | TACGCGATGAGGCTTATTAAG | N/A |
| TDA1-K68R-3-1213 | CTTAATAAGCCTCATCGCGTA | N/A |
| TDA1-S483T484A-5-2436 | TGAAAAAGAATGCTGCTTTTGTC | N/A |
| TDA1-S483T484A-3-2458 | GACAAAAGCAGCATTCTTTTTCA | N/A |
| TDA1-S393A-5-2162 | AGGTCACCTTCCAAAGCAAGA | N/A |
| TDA1-S393A-3-2182 | TCTTGCTTTGGAAGGTGACCT | N/A |
| TDA1-S509A-5-2518 | AATCGCGCGCGAATTTCAAT | N/A |
| TDA1-S509A-3-2537 | ATTGAAATTCGCGCGCGATT | N/A |
| TDA1-T513A-5-2518 | ATCGCGGTCAAATTTCAATGCCCCCAAAAC | N/A |
| TDA1-T513A-3-2547 | GTTTTGGGGGCATTGAAATTTGACCGCGAT | N/A |
| TDA1-5-BamHI-301 | AAGGTTGGATCCGGTTTAAGATGCTCTTTTACCAAT | N/A |
| TDA1-3-Kpn1-3150 | AGTTCCGGTACCGTAAGCTAAATGATCACTGGGCTG | N/A |
| TDA1-5-Nde1-1001 | AAGGTTCATATGACTACAGCTAGTTCTTCTGCT | N/A |
| TDA1-3-FLAG-BamH1-2758 | AGTTCCGGATCCTTGTCATCGTCGTCCTTGTAGTCTATCTGCATGGTTTCATC | N/A |
| SNF1-pKL-5-901 | GCTTTTGATAATATATAGAGGCTGTTTCAATAATCATAGCGAAAGAAATATAACCCGGGTTAATTAAGGC | N/A |
| SNF1-pKL-3-2951 | GATACATAAAAAAAAGGGAACTTCCATATCATTCTTTTACGTTCCACCATATCGATGAATTCGAGCTCG | N/A |
| SNF1-pKL-5-2850 | ATTTAACAACAAAACTAATTATGGAATTAGCCGTTAACAGTCAAAGCAATCGTACGCTGCAGGTCGAC | N/A |
| Snf1-3-3350 | TGATTTGATATCGCTTCACA | N/A |
| SNF1-K84R-5-1228 | TACGGGTCAAAAAGTTGCTCTAAGAATCAT | N/A |
| SNF1-K84R-3-1529 | ATGATTCTTAGAGCAACTTTTTGACCCGTA | N/A |
| Snf1-5-Spe1-301 | AGTTCCACTAGTATACGGTTATAAACTGTACA | N/A |
| Snf1-3-Sal1-1660 | AAGGTTGTCGACAGGAGCCGCATAATTGGGAG | N/A |
| SNF1-pFU-5-2849 | ATTTAACAACAAAACTAATTATGGAATTAGCCGTTAACAGTCAAAGCAATTTCATCTCCGGTTCTGCTGCTAGA | N/A |
| SNF1-pFU-3-2951 | GATACATAAAAAAAAGGGAACTTCCATATCATTCTTTTACGTTCCACCATCCTCGAGGCCAGAAGACTAAGAGG | N/A |
| Rim11-pKL-5-907 | GACGTCTGATTCACACATTACTGGGCAAGATCTTGACATAGCATTACATT-CGTACGCTGCAGGTCGAC | N/A |
| RIM11-pKL-5-2061 | CTCCCGATGAACTATCATCTGTAAAAAAAAAGCTATATCCGAAGTCTAAGCGTACGCTGCAGGTCGAC | N/A |
| RIM11-pKL-3-2163 | GTTCCTTCCTTCTCCCATTATTCTTGCCTGGGCTCCCTCCGGTGCTATCAATCGATGAATTCGAGCTCG | N/A |
| RIM11-pFU-5-2061 | CTCCCGATGAACTATCATCTGTAAAAAAAAAGCTATATCCGAAGTCTAAGTTCATCTCCGGTTCTGCTGCTAGA | N/A |
| RIM11-pFU-3-2163 | GTTCCTTCCTTCTCCCATTATTCTTGCCTGGGCTCCCTCCGGTGCTATCACCTCGAGGCCAGAAGACTAAGAGG | N/A |
| RIM11-3-2515 | AACACCCACTACATCAGCTC | N/A |
| RIM11-5-K68R-1193 | GTTGCTATTAGGAAAGTCCT | N/A |
| RIM11-3-K68R-1212 | AGGACTTTCCTAATAGCAAC | N/A |
| RIM11-Nhe1-5-401 | AAGGTTGCTAGCGTGCTGTTATCGTTGTGTTG | N/A |
| RIM11-Xho1-3-1866 | AGTTCCCTCGAGTTGAAAACACGTGACAATGGTATT | N/A |
| MCK1-pFU-5-901 | TATTGATAGGAGTTAAGCCCAAGACTACAGAGTTCTTTGCTTCATCTTTCTTCATCTCCGGTTCTGCTGCTAGA | N/A |

| Reagent/resource | Reference or source | Identifier or catalog number |
|---|---|---|
| MCK1-pFU-3-2178 | TTGTTCATTAAATTTTCCGAGGGGAAAGAGAACAAATTAATAGAAAATTACCTCGAGGCCAGAAGACTAAGAGG | N/A |
| MCK1-5-851 | GTGAGTGCCATTGTGTCGTG | N/A |
| MCK1-3-2216 | ATAAACAGCGGATCAAAGGT | N/A |
| HOG1-pKL-5-901 | AACCTTATTTATTTTCTCTTTCTTCTATATTGGTAAATACTAGACTCGAATAACCCGGGTTAATTAAGGC | N/A |
| HOG1-pKL-3-2408 | CTATACAACTATATACGTAAATACTTTTATGAGTACCATAAAAAAAAGAAATCGATGAATTCGAGCTCG | N/A |
| HOG1-5-838 | TGTTTGTATAGTGGAAGAGG | N/A |
| HOG1-3-2507 | TACAAGAAAATCCAATGCGG | N/A |
| FUS3-5-pFU-800 | CTGCATTCTCTCAGATTTTAGATGATGCGGTTTTTTACAGGGCATTGAAATTCATCTCCGGTTCTGCTGCTAGA | N/A |
| FUS3-3-pFU-2261 | AATTAACAGCCGCCGACGGCCCGTGGCGCGCGATCACGGAGTTGCGTAACCCTCGAGGCCAGAAGACTAAGAGG | N/A |
| FUS3-5-570 | TTGGTGTTCATTCAAGGGGC | N/A |
| FUS3-3-2547 | GGGGTAGAACACATCTAACG | N/A |
| KSS1-5-pFU-800 | TGAACAGTTACATATTGTGCTTTGCAGTCGTTAAATTTCCCGAACTGTTTTTCATCTCCGGTTCTGCTGCTAGA | N/A |
| KSS1-3-pFU-2307 | TTTCGTGTACTTCTGCTGGCGAAAAAAAAAAAACTTGTTGCCGTGGCGCCACCTCGAGGCCAGAAGACTAAGAGG | N/A |
| KSS1-5-565 | TGAAGCCTCAGATTACCTTG | N/A |
| KSS1-3-2555 | ACGAATGCGAACATCTAGCC | N/A |
| SLT2-5-pFU-800 | CAGGAATATATCGAAAAAAAAAGTGAGGGAAATCAGATCCTACACAAATATTCATCTCCGGTTCTGCTGCTAGA | N/A |
| SLT2-3-pFU-2655 | GAATTCAAGAGGCGATAACAAACTTCCGCGGAGTACGATTAAGATAAGCGCCTCGAGGCCAGAAGACTAAGAGG | N/A |
| SLT2-5-604 | GGTTCGAATACTTGTGAGCC | N/A |
| SLT2-3-2847 | CCCATAAAGGGCTTCTCAGT | N/A |
| ACT1-5-1371 | TTGCCGGTGACGACGCTCCT | N/A |
| ACT1-3-1670 | GAGTCATCTTTTCTCTGTTT | N/A |
| FBP1-5-1241 | CAAAAGAAGTTGGACGTTCT | N/A |
| FBP1-3-1620 | TGAGTCAAGATGAATTCGCC | N/A |
| PCK1-5-1200 | AAACCGGAAGATCTCCAAAG | N/A |
| PCK1-3-1500 | TCAGGTTCTCCAAAATGGGC | N/A |
| CIT3-5-1201 | GATGAGAGGTAATCAGTCAA | N/A |
| CIT3-3-1500 | GCAAGTTGGGTCATAGGGTG | N/A |
| **Chemicals, enzymes and other reagents** | | |
| D-glucose | FUJIFILM Wako Pure Chemical | 043-31163 |
| HIPOLYPEPTON | SHIOTANI M.S. | 399-02267 |
| Yeast Extract Dried | Nacalai Tesque | 15838-45 |
| Yeast Nitrogen Base w/o Amino Acids | ForMedium | CYN0210 |
| EDTA-free complete protease inhibitor cocktail | Roche | C762Q78 |
| EDTA-free phosphatase inhibitor cocktail | Nacalai Tesque | 07575-51 |
| Triton X-100 | Sigma-Aldrich | 30-5140-5 |
| Ab-Capcher MAG2 | Protenova | P-052-10 |
| $Ni^{2+}$-agarose beads | FUJIFILM Wako Pure Chemical | 143-09763 |
| Phos-tag acrylamide | FUJIFILM Wako Pure Chemical | 304-93526 |
| 5-Fluoroorotic Acid Monohydrate | FUJIFILM Wako Pure Chemical | 064-03664 |

| Reagent/resource | Reference or source | Identifier or catalog number |
|---|---|---|
| Luminata Forte Western HRP Substrate system | Merck Millipore | WBLUM0100 |
| Chemi-Lumi One L system | Nacalai Tesque | 07880-54 |
| TriPure Isolation Reagent | Roche | 11667157001 |
| ReverTra Ace® qPCR RT Master Mix with gDNA Remover | TOYOBO | FSQ-301 |
| THUNDERBIRD® Next SYBR™ qPCR Mix | TOYOBO | QPX-201 |
| NucleoSpin® RNA Plus kit | MACHEREY-NAGEL | 740990.5 |
| 1NM-PP1 | Cayman Chemical | 221244-14-0 |
| DMSO | FUJIFILM Wako Pure Chemical | 043-07216 |
| **Software** | | |
| Fiji | https://imagej.net/software/fiji/ | N/A |
| GraphPad Prism 9.0 | https://www.graphpad.com/ | N/A |
| Xcalibur 4.1.50 | Thermo Fisher Scientific | N/A |
| Proteome Discoverer 2.4.1.15 | Thermo Fisher Scientific | N/A |
| SEQUEST | Thermo Fisher Scientific | N/A |
| **Other** | | |
| Micro Smash MS-100R | TOMY SEIKO | N/A |
| LAS4000 mini | GE Healthcare Biosciences | N/A |
| HPLC-MS/MS system | AMR | N/A |
| NTCC-360/100-3-125 | Nikkyo Technos | N/A |

## Methods and protocols

### Yeast strains and growth conditions

Strains used in this study are listed in Reagents and tools table. The strains were grown in either YPD (1% yeast extract, 2% peptone, and 2% D-glucose) or SD standard minimal medium (0.67% yeast nitrogen base without amino acids, 2% glucose, and all standard amino acids (drop out mix). Where indicated, 0.05% glucose, 3% glycerol or 2% acetate instead of 2% glucose were used as the carbon source.

### Genetic manipulation

Gene disruption was performed by replacing the entire coding sequence of the genes with a marker gene via homologous recombination. Chromosome fusions of HA or FLAG to the 3′-terminus of the gene were conducted using PCR-based gene disruption and modification. Roughly, the sequence containing the tag-encoding gene (HA or FLAG), the *ADH1* terminator, and a marker gene was amplified using PCR from pFU562 (for HA-tagging) or pKL259 (for FLAG-tagging) with a primer set containing the homologous region (50 mer) of each gene. The PCR-amplified fragments were directly inserted into the chromosome via homologous recombination. Serine or threonine-to-alanine substitution mutants and kinase-dead mutants were generated as follows. The mutated gene sequences were cloned into the pRS306 integration vector (Sikorski and Hieter, 1989). The resulting plasmids were digested with appropriate restriction enzymes (*SnaB* 1 for Hxk2-S15A; *Sal* I for Tda1-S393A, Tda1-S483A/T484A, Tda1-S509A, Tda1-T513A, Tda1-S483A/T484A/S509A, and Tda1-S483A/T484A/T513A; *Swa* I for Tda1-K68R; *Bgl* II for Snf1-K84R; and *Spe* I for Rim11-K68R), and the resulting linearized DNA was used for homologous recombination. Subsequently, the mutants were cultured on synthetic medium containing 1 mg/ml 5-Fluoroorotic acid, and those lacking the *URA3* gene were selected. Successful genetic manipulation were confirmed via genomic PCR and/or immunoblot.

### Antibodies

Polyclonal antibodies to Tda1, Hxk2, Sic1, and Cdc48 were generated in rabbits against fragments of recombinant Tda1 (residues 289–586), Hxk2 (residues 16–165), Sic1 (residues 1–284), and Cdc48 (residues 1–218) as antigens. The rabbit

polyclonal antibodies to hemagglutinin (HA) and Myc were raised in our laboratory. The mouse monoclonal antibody (mAb) to HA (12CA5) and FLAG (M2) were obtained from Roche (Mannheim, Germany) and Sigma-Aldrich (St. Louis, USA), respectively.

### Plasmid construction

The WT and S15A mutant forms of HXK2, tagged with Myc at their C-termini, were subcloned into the pET23b vector (Novagen, Madison, USA). TDA1, tagged with FLAG at the C-terminus, was also subcloned into the pET23b vector.

### Yeast extracts

Cells used for extraction were killed by adding TCA directly to the medium at a final concentration of 10% at the time of collection. Total cell extracts were prepared using the trichloroacetic acid (TCA) lysis method. Briefly, cells were resuspended in 20% TCA and lysed by vigorous vortexing for 10 min. A fourfold volume of 5% TCA was subsequently added, and the extracts were centrifuged at $20,000 \times g$ for 2 min at room temperature. The pellets were washed with ice-cold acetone and then resuspended in SDS-PAGE sample buffer.

### Immunoblot analysis

Proteins were separated by Phos-tag SDS-PAGE (FUJIFILM Wako Pure Chemical, Osaka, Japan) or SDS-PAGE and transferred onto Immobilon-P membranes (Millipore, Billerica, USA). To detect the in vivo and in vitro phosphorylation of Hxk2 and the in vitro phosphorylation of Tda1, Phos-tag gels, containing 8% acrylamide, 100 μM Phos-tag acrylamide (FUJIFILM Wako Pure Chemical), and 100 μM MnCl$_2$, were used. The membranes were incubated with the following antibodies: anti-Hxk2 (1:1500 dilution; produced in our lab), anti-Tda1 (1:500 dilution; produced in our lab), anti-Sic1 (1:500 dilution; produced in our lab), anti-Cdc48 (1:200,000 dilution; produced in our lab), anti-HA (1:10,000 dilution; produced in our lab), anti-Myc (1:10,000 dilution; produced in our lab), and HRP-conjugated anti-FLAG antibody (1:10,000 dilution; Sigma-Aldrich). HRP-conjugated anti-rabbit IgG (1:5000 dilution; Sigma-Aldrich) was used as the secondary antibody. Immunodetection was performed using the Luminata Forte Western HRP Substrate system (Merck Millipore, Burlington, USA) or the Chemi-Lumi One L system (Nacalai Tesque, Kyoto, Japan), and signals were visualized using a bioanalyzer (LAS4000 mini; GE Healthcare Biosciences, Piscataway, USA).

### Immunoprecipitation analysis

Cells were harvested and resuspended in a buffer containing 40 mM Tris-HCl (pH 7.5) and 150 mM NaCl, supplemented with 1× EDTA-free complete protease inhibitor cocktail (Roche) and 1× EDTA-free phosphatase inhibitor cocktail (Nacalai Tesque). The cells were disrupted using glass beads and a Micro Smash MS-100R (TOMY SEIKO, Tokyo, Japan). Proteins were solubilized with 0.5% Triton X-100 for 5 min at 4 °C. After centrifugation at $20,000 \times g$ for 10 min at 4 °C to remove debris and unsolubilized material, the supernatant was incubated with anti-HA antibody (12CA5, Sigma-Aldrich) for 40 min at 4 °C with rotation. Ab-Capcher MAG2 (Protenova, Kagawa, Japan) was then added, and samples were rotated for an additional 15 min at 4 °C. The beads were washed three times with a solution containing 40 mM Tris-HCl (pH 7.5), 150 mM NaCl, and 0.5% Triton X-100. Immunoprecipitated

proteins were eluted by incubating the beads in SDS-sample buffer for 5 min at 88 °C, followed by immunoblot analysis.

### In vitro phosphorylation assay

Hxk2-Myc-His$_6$, Hxk2-S15A-Myc-His$_6$, and Tda1-FLAG-His$_6$ were expressed in *E. coli* BL21 (DE3) and purified using Ni$^{2+}$-agarose beads (FUJIFILM Wako Pure Chemical). Tda1-3×HA, Snf1-3×HA, and Rim11-3×HA were purified from yeast cells grown in YP with 0.05% glucose medium via immunoprecipitation with anti-HA antibody (12CA5; Sigma-Aldrich) and Ab-Capcher MAG2 beads (Protenova, Kagawa, Japan). For the in vitro Hxk2 phosphorylation assay, 50 ng of Hxk2-WT-Myc-His$_6$ or Hxk2-S15A-Myc-His$_6$ was incubated with beads bound to C-terminally 3xHA-tagged Tda1, Snf1, or Rim11 in a 10 μL solution containing 40 mM Tris-HCl (pH 7.5), 60 mM NaCl, 5 mM MgCl$_2$, and 1.5 mM ATP at 30 °C for 1 h. For the in vitro Tda1 phosphorylation assay, 50 ng of Tda1-FLAG-His$_6$ was incubated with beads bound to C-terminally 3xHA-tagged Snf1 under the same conditions as the Hxk2 phosphorylation assay.

### Protein mass spectrometry

Immunoprecipitated proteins were partially separated (~1 cm) via SDS-PAGE. In-gel digestion was carried out following the protocol established by Kano et al (Kano et al, 2023). For analysis, nano-flow reverse-phase liquid chromatography coupled with tandem mass spectrometry (MS) was utilized. A capillary reverse-phase HPLC-MS/MS system consisted of a Dionex U3000 gradient pump equipped with a VICI Cheminert valve and Q Exactive equipped with a nano-electrospray ionization (NSI) source (AMR, Tokyo, Japan). The desalted peptides were introduced into a C18 reverse-phase capillary column (NTCC-360/100-3-125, 125×0.1 mm, Nik-kyo Technos, Tokyo, Japan) for separation. Data acquisition for peptide spectra, spanning an m/z range of 350–1800, was handled by the Xcalibur 4.1.50 software (Thermo Fisher Scientific, Waltham, USA). Successive MS scans were followed by ten data-dependent high-energy collision dissociation (HCD) MS/MS scans, targeting the ten most intense precursor ions. MS/MS spectra were processed and peak lists were generated using Proteome Discoverer version 2.4.1.15 (Thermo Fisher Scientific). SEQUEST (Thermo Fisher Scientific) was employed to search against the Saccharomyces cerevisiae peptide sequences from SwissProt (TaxID = 4932). The search parameters included the selection of the enzyme, with allowance for up to two missed cleavage sites, a peptide mass tolerance of 10 ppm, and an MS/MS tolerance of 0.02 Da. Modifications were specified as follows: carbamidomethylation of cysteine as a fixed modification, and oxidation (M), phosphorylation (S, T, Y), and ubiquitination (K) as variable modifications. Peptide identification was based on a significant Xcorr value, applying a high-confidence filter. Identified peptides and their modifications from SEQUEST were manually reviewed and filtered to create a verified list of peptide identifications and modifications derived from HCD MS/MS data.

### RNA extraction and RT-qPCR

Cells were grown to log phase in YPD medium at 30 °C, followed by an additional 6-hour incubation in YP with 3% glycerol medium. After harvesting, the cells were lysed using glass beads and a Micro Smash™ MS-100R (TOMY). Total RNA was extracted using the TriPure Isolation Reagent (Roche) following the manufacturer's

instructions, and reverse transcription was performed with ReverTra Ace® qPCR RT Master Mix with gDNA Remover (TOYOBO, Osaka, Japan). Quantitative PCR was conducted in triplicate using THUNDERBIRD® Next SYBR™ qPCR Mix (TOYOBO). Gene expression levels were normalized to *ACT1*. The primer sequences for amplifying the ORF region are listed in Reagents and tools table.

### RNA-seq

Cells were cultivated to the log phase in YPD medium at 30 °C and then further cultured in YP medium with 3% glycerol for 6 h. Total RNA was isolated using the NucleoSpin® RNA Plus kit (MACHEREY-NAGEL, North Rhine-Westphalia) according to the manufacturer's protocol. One μg of total RNA was used for mRNA purification using NEBNext Oligo d(T)25 beads (NEBNext poly(A) mRNA Magnetic Isolation Module; New England Biolabs, MA, USA). First-strand cDNA synthesis was then carried out using the NEBNext Ultra II RNA Library Prep Kit for Illumina (New England Biolabs), alongside NEBNext Multiplex Oligos for Illumina (New England Biolabs) following the manufacturer's guidelines. Paired-end 81-bp sequencing on the Illumina NextSeq 550 platform (Illumina, CA, USA) yielded from 4,250,462 to 5,974,049 reads (Appendix Table S2).

### RNA-seq data analysis

Yeast genome assembly and annotation (accession: GCA_000146045.2) was downloaded from the NCBI genome database. Raw sequencing reads with Phred quality scores lower than 15 or lengths shorter than 15 bp were filtered out using fastp preprocessor (v0.23.2). The remaining reads were aligned to the yeast reference genome using HISAT2 (v2.2.1). Quantification of the aligned reads was performed using RSEM (v1.3.1). The resulting expected read counts were count per million (CPM) normalized and log2 transformed. Hierarchical clustered heatmap and principal component analysis (PCA) were visualized using iDEP v.0.96 web-server. Log2 transformed fold changes for each sample pair were applied to generate scatter plots. The gene lists were obtained from a previous study (Oh et al, 2018).

### Statistical analysis

Western blot images were quantified using Fiji (NIH). Statistical analyses were conducted with GraphPad Prism 9, employing two-tailed unpaired *t* tests or one-way ANOVA followed by Dunnett's multiple comparison test.

## Data availability

The RNA-seq data are deposited in NCBI GEO under the accession GSE290495. The data can be accessed at the following page: https://www.ncbi.nlm.nih.gov/geo/query/acc.cgi?acc=GSE290495. The mass spectrometry data are deposited in jPOST under PXID: PXD060707, and this data can be accessed at the following page: https://repository.jpostdb.org/preview/160356263067ac27880946a.

The source data of this paper are collected in the following database record: biostudies:S-SCDT-10_1038-S44319-025-00456-y.

## Peer review information

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

## Acknowledgements

We would like to thank K, Kamura for technical assistance. This work was supported by JSPS KAKENHI Grant Number 2522K06141 to KO, 2522K05558 to KN and 2523K27126 to TK.

## Author contributions

**Kazuki Nonaka**: Data curation; Formal analysis; Investigation; Writing—original draft; Writing—review and editing. **Kohei Nishimura**: Conceptualization; Funding acquisition; Project administration. **Kazuma Uesaka**: Software; Formal analysis; Methodology. **Emi Mishiro-Sato**: Software; Formal analysis; Methodology. **Minako Fukase**: Formal analysis; Investigation. **Rei Kato**: Investigation. **Fumihiko Okumura**: Conceptualization; Project administration. **Kunio Nakatsukasa**: Conceptualization; Project administration. **Keisuke Obara**: Conceptualization; Funding acquisition; Project administration.

**Takumi Kamura**: Conceptualization; Resources; Data curation; Supervision; Funding acquisition; Investigation; Methodology; Writing—original draft; Project administration; Writing—review and editing.

Source data underlying figure panels in this paper may have individual authorship assigned. Where available, figure panel/source data authorship is listed in the following database record: biostudies:S-SCDT-10_1038-S44319-025-00456-y.

## Disclosure and competing interests statement

The authors declare no competing interests.

