## [Peer Review File · EMBO Reports]

Snf1 and yeast GSK3- β activates Tda1 to suppress glucose starvation signaling

Kazuki Nonaka, Kohei Nishimura, Kazuma Uesaka, Emi Mishiro-Sato, Minako Fukase, Rei Kato, Fumihiko Okumura, Kunio Nakatsukasa, Keisuke Obara, and Takumi Kamura

Corresponding author(s): Takumi Kamura (kamura.takumi.k1@f.mail.nagoya-u.ac.jp) , Keisuke Obara (obara.keisuke.r2@f.mail.nagoya-u.ac.jp), Kohei Nishimura (Nishimura.kohei.x8@f.mail.nagoya-u.ac.jp)

Review Timeline:

Submission Date:	24th Oct 24
Editorial Decision:	20th Nov 24
Revision Received:	19th Feb 25
Editorial Decision:	20th Mar 25
Revision Received:	24th Mar 25
Accepted:	3rd Apr 25

Editor: Deniz Senyilmaz Tiebe

Transaction Report:

Dear Prof. Kamura,

Thank you for submitting your research manuscript to our journal, which was now seen by three referees, whose reports are copied below.

Referees express interest in the proposed mechanism by which Tda1 regulates the glucose starvation signaling in yeast. However, they also raise some concerns that need to be addressed to consider publication here.

I find the reports informed and constructive, and believe that addressing the concerns raised will significantly strengthen the manuscript. As the reports are below, and I think all points need to be addressed, I will not detail them here. Please contact me if you have questions or comments regarding the revision for further discussion (also by video chat).

Given these positive recommendations, we would like to invite you to submit a revised manuscript. Please revise your manuscript with the understanding that the referee concerns (as in their reports) must be fully addressed and their suggestions taken on board. Please address all referee concerns in a complete point-by-point response. Acceptance of the manuscript will depend on a positive outcome of a second round of review. It is EMBO reports policy to allow a single round of major experimental revision only and acceptance or rejection of the manuscript will therefore depend on the completeness of your responses included in the next, final version of the manuscript.

We realize that it is difficult to revise to a specific deadline. In the interest of protecting the conceptual advance provided by the work, we recommend a revision within 3 months. Please discuss the revision progress ahead of this time with me if you require more time to complete the revisions, or if you have questions or comments regarding the revision (also by video chat).

1. A data availability section providing access to data deposited in public databases is missing (where applicable).
2. Your manuscript contains statistics and error bars based on $n=2$. Please use scatter plots in these cases.

You can submit the revision either as a Scientific Report or as a Research Article. For Scientific Reports, the revised manuscript can contain up to 5 main figures and 5 Expanded View figures, and it should not exceed 27000 characters. If the revision leads to a manuscript with more than 5 main figures it will be published as a Research Article. In this case the Results and Discussion section should be separate. If a Scientific Report is submitted, these sections have to be combined. This will help to shorten the manuscript text by eliminating some redundancy that is inevitable when discussing the same experiments twice. In either case, all materials and methods should be included in the main manuscript file.

4) a .docx formatted letter INCLUDING the reviewers' reports and your detailed point-by-point responses to their comments. As part of the EMBO publication's Transparent Editorial Process, EMBO reports publishes online a Review Process File (RPF) to accompany accepted manuscripts. This File will be published in conjunction with your paper and will include the referee reports,

your point-by-point response and all pertinent correspondence relating to the manuscript.

<https://www.embopress.org/page/journal/14693178/authorguide#transparentprocess>

5) a complete author checklist, which you can download from our author guidelines

<https://www.embopress.org/page/journal/14693178/authorguide>. Please insert information in the checklist that is also reflected in the manuscript. The completed author checklist will also be part of the RPF.

6) Please note that all corresponding authors are required to supply an ORCID ID for their name upon submission of a revised manuscript (<<https://orcid.org/>>). Please find instructions on how to link your ORCID ID to your account in our manuscript tracking system in our Author guidelines

<<https://www.embopress.org/page/journal/14693178/authorguide#authorshipguidelines>>

7) Before submitting your revision, primary datasets produced in this study need to be deposited in an appropriate public database (see <https://www.embopress.org/page/journal/14693178/authorguide#datadeposition>). Please remember to provide a reviewer password if the datasets are not yet public. The accession numbers and database should be listed in a formal "Data Availability" section placed after Materials & Method (see also

<https://www.embopress.org/page/journal/14693178/authorguide#datadeposition>). Please note that the Data Availability Section is restricted to new primary data that are part of this study. * Note - All links should resolve to a page where the data can be accessed. *

Additional information on source data and instruction on how to label the files are available:

<https://www.embopress.org/page/journal/14693178/authorguide#sourcedata>

9) Our journal encourages inclusion of *data citations in the reference list* to directly cite datasets that were re-used and obtained from public databases. Data citations in the article text are distinct from normal bibliographical citations and should directly link to the database records from which the data can be accessed. In the main text, data citations are formatted as follows: "Data ref: Smith et al, 2001" or "Data ref: NCBI Sequence Read Archive PRJNA342805, 2017". In the Reference list, data citations must be labeled with "[DATASET]". A data reference must provide the database name, accession number/identifiers and a resolvable link to the landing page from which the data can be accessed at the end of the reference. Further instructions are available at <http://www.embopress.org/page/journal/14693178/authorguide#referencesformat>

10) Regarding data quantification (see Figure Legends:

<https://www.embopress.org/page/journal/14693178/authorguide#figureformat>)

11) The journal requires a statement specifying whether or not authors have competing interests (defined as all potential or actual interests that could be perceived to influence the presentation or interpretation of an article). In case of competing interests, this must be specified in your disclosure statement. Further information: <https://www.embopress.org/competing->

interests

12) Please also note our reference format:

13) All Materials and Methods need to be described in the main text using our 'Structured Methods' format, which is required for all research articles. According to this format, the Methods section includes a Reagents and Tools Table (listing key reagents, experimental models, software and relevant equipment and including their sources and relevant identifiers) followed by a Methods and Protocols section describing the methods using a step-by-step protocol format. The aim is to facilitate adoption of the methodologies across labs. More information on how to adhere to this format as well as a downloadable template (.docx) for the Reagents and Tools Table can be found in our author guidelines:

I look forward to seeing a revised version of your manuscript when it is ready. Please let me know if you have questions or comments regarding the revision.

Kind regards,

Deniz Senyilmaz Tiebe

Deniz Senyilmaz Tiebe, PhD
Senior Scientific Editor
EMBO Reports

Referee #1:

This paper by Nonaka et al. is a great study which extends our knowledge on the regulation of hexokinase phosphorylation in yeast, and more globally, on how yeast cells react to glucose starvation.

Background and significance.

Hexokinase 2 has long been described as carrying two functions, both as a metabolic enzyme and a glucose signaling regulator. Various hypotheses over the years have been made to explain these functions, notably after the discovery of a nuclear localization of Hxk2 and its phosphorylation on Ser15.

Whereas the published data on potential nuclear functions of Hxk2 is still a bit shady, Ser15 phosphorylation was clearly established as preventing Hxk2 dimerization, and Hxk2 monomers have an increased affinity towards glucose.

Past studies clearly established that Hxk2-Ser15 is phosphorylated by Tda1, a NUA1 orthologue.

Additional work revealed that Tda1 requires its prior phosphorylation to become active, and that the AMPK orthologue Snf1 is partially in charge of this activation by phosphorylation of Tda1 on Ser483/Thr484.

In this study, Nonaka et al. reveal the identity of a second kinase, the yeast glycogen synthase kinase 3 (GSK-3), notably Rim11 and Mck1, on the phosphorylation of Tda1-Ser509 which is parallel to that of Snf1 on Ser483/Thr484 and is required for the full activity of Tda1.

Moreover, this paper provides evidence that the deletion of TDA1 or the lack of phosphorylation of Hxk2-Ser15 cause similar glucose-related phenotypes (gene expression and resistance to glucose starvation).

These findings unambiguously enhance our understanding of glucose sensing mechanisms and provide a careful analysis of the function of new actors of this pathway.

Summary.

The authors generated specific antibodies against Tda1 and Hxk2 that are great tools in combination with PhosTag gels to study their phosphorylation. Previous data on the role of glucose availability, Snf1 or Tda1 in Hxk2 phosphorylation are nicely confirmed and the data is quantified.

The authors developed a nice in vitro assay in which they can reconstitute both Hxk2 and Tda1 phosphorylation from kinases immunopurified from yeast. The controls are accurate. The effects of phosphosite point mutations also recapitulate the results obtained on the kinase mutants. Altogether, the results indicate that 1/ Tda1 activity and 2/ its full phosphorylation requires Snf1, but that another kinase also contributes to this.

After Tda1 IP + MS, the authors find many kinases associated to Tda1 and focus on the GSK-3 homologues Rim11 and Mck1, based on previously published data (proteomics and genetics). They find that the double mutant rim11 Δ mck1 Δ displays defects

in both Tda1 phosphorylation and consequently, Hxk2 phosphorylation. They also recapitulate these findings by in vitro phosphorylation.

The authors identify the GSK-3 dependent phosphosites on Tda1 and show that Ser509 is the primary site and is functionally linked to Tda1 activity. The importance of a prior phosphorylation on Thr513 is also noted, although the kinase responsible could not be identified.

Altogether, this clearly shows that Tda1 is activated by phosphorylation by 2 kinases at two different sites, and that these pathways work in parallel and in an additive manner for a full activation of Tda1. A third kinase is likely to prime the phosphorylation of Tda1 by GSK-3 on Ser509.

A very interesting aspect of the study is also that it paves the way for future work on the functional consequences of Hxk2 phosphorylation on glucose signaling. Indeed, despite heavy work on the phosphorylation of Hxk2, no clear functional consequences were ever found except the effect on dimerization. Similarly, the relationship between Tda1 and glucose signaling remained elusive.

Although this is a bit fragile, the results suggest that TDA1 deletion or HXK2-S15A mutation cause a better growth on the non-fermentable carbon source Glycerol. Also, the double mutant rim11Δ mck1Δ displays growth defects in glycerol medium. However, the authors make the interesting discovery that TDA1 deletion or HXK2-S15A mutation lead to a similar transcriptional reprogramming, which overall suggests they are deficient in glucose repression. This leads to many questions which are probably out of the scope of this manuscript for the moment, but simple experiments are suggested that would help give better clues on how Tda1 controls the glucose sensing pathway.

Suggestions are made to improve the discussion, too.

Major points.

- Introduction, l. 54, the authors state "the gamma subunit detects a drop in the cellular (AMP+ADP)/ATP ratio" but it is the opposite- either a drop in ATP/(AMP+ADP) or an increase in (AMP+ADP)/ATP.
- Could the authors discuss the fact that in the snf1Δ strain, there is more phosphorylation of Hxk2 to begin with?
- Could the authors quantify the growth differences shown by drop tests in Fig 6A using growth curves. It is an important aspect of the paper and it would be nice to have robust data at this stage. Is this restricted to glycerol, or also observed for other C sources such as raffinose or galactose? Also, how is the HXK2-S15A generated? Is it a clean replacement at the endogenous locus, how was this made? Please include details in Material and methods section.
- Same question for all the Ser/Thr mutants generated for TDA1.
- RNA-seq - are the three columns for each mutants biological replicates? What does the colour code correspond to? (What is the reference? WT in Glc medium?). How was clustering made? Are all genes represented? Please provide more information about this experiment in the methods/legend/text sections, as it seems a bit expedited.
- Fig 6C and 6D. I am not sure about the interpretation of these graphs. Why plot the ratios (xxx/WT) against each other rather than the values themselves? If the values on X and Y axes are divided by the same WT value then it should be the same. What is the coefficient indicated (regression, correlation?) Globally it seems that tda1Δ phenocopies Hxk2-S15A and is anti-correlated with snf1Δ but I feel that these graphs do not show this so clearly. Would principal component analysis display a better view of the results?
- I was wondering if the authors had tried to look at the growth of a snf1Δ tda1Δ or snf1Δ Hxk2-S15A on glycerol, to know whether the lack of Hxk2 phosphorylation would restore the expression of glucose-repressed genes in the snf1Δ, or on the opposite, whether Snf1 activity is required for the tda1Δ or Hxk2-S15A phenotypes? These simple experiments could help pinpoint whether tda1Δ mimics a glucose derepressed mutant like reg1Δ.
- I think the discussion is a bit long, it could be edited to be more straight-to-the-point and avoid redundancies, but also completed. What do we know about the relationships between glucose signaling / metabolism and Rim11/Mck1 (echoing to the data presented in 6A which suggests a lack of glucose repression?). How would the glucose signal be perceived by GSK-3, and/or what could be the mechanism by which Hxk2 becomes suddenly phosphorylated by GSK-3? (eg. the priming site Thr513, for example). Overall, can the authors put forward hypotheses on how Hxk2 phosphorylation controls gene expression? For example data in the literature shows that Hxk2 monomers and dimers have a different affinity towards glucose. Also, data by Lesko et al 2023 (doi:10.1371/journal.pgen.1010745) suggest that hxk2Δ does not confer major transcriptional changes. How the authors reconcile these data with their data on the Hxk2-SA mutant? This should be discussed. Finally, I think there is a paradox that is not discussed. Snf1 is a positive regulator of Tda1 activity, yet the transcriptomics profile are opposite to each other. Maybe this could be mentioned and discussed.

Minor points.

- the authors state "when glucose levels drop, more efficient energy production pathways are activated" (l.71) - I'm not sure about the word "efficient" here, yeast cells indeed trigger the onset of respiration but they rather turn to other carbon sources. In a way it is more efficient because it is better adapted, but "efficient" could also be read as a better output in ATP and I'm not sure this is the case
- The authors refer to past studies by the Moreno lab on the potential nuclear functions of Hxk2. This work has been criticized (Kriegel et al. 2016; doi:10.1074/jbc.L116.735514) and invalidated (Lesko et al 2023; doi:10.1371/journal.pgen.1010745). Same thing for the data suggesting that Snf1 is the Hxk2 kinase (Fernandez-Garcia et al., 2012, doi: 10.1074/jbc.M112.401679), which was not confirmed in subsequent studies (Kaps et al. 2015; doi:10.1074/jbc.M114.595074). While I agree that these studies could be mentioned, I think that now is the time to start downplaying the importance of this work to clear up the field, by being even more cautious about these conclusions when mentioned in the text and/or limit citations. But of course this is at the

author's discretion.

- I believe the authors should mention Mig1 in the introduction (at the same time as Adr1 etc, l. 77)

- l.103, "GSK-3 is constitutively active but can be inactivated" - then I suppose that it is not constitutively active? Do the authors mean that it is always active in rich medium?

-Typo in Fig 6D, "TCA cycles"

Referee #2:

Review of "Tda1, activated by Snf1 and yeast GSK-3, negatively regulates the glucose starvation signaling"

The paper presents strong data showing that both Snf1 and GSK3 kinases activate the Tda1 kinase which in turn phosphorylates Hxk2 on its Serine 15 residue, mediating the nuclear export of Hxk2. Snf1 phosphorylates Tda1 at S483 and T484, whilst GSK3 phosphorylates Tda1 at S509. Unexpectedly, deletion of Tda1 enhances respiratory growth and gene expression, which is in contrast to the body of literature about Hxk2 localization. Some more experiments showing how these mutations in Tda1 affect Hxk2 localization would be highly beneficial.

General comments-Major

1) The title would be clearer if it were: "Independent activation of Tda1 by Snf1 and GSK3 negatively regulates glucose starvation signaling in *S. cerevisiae*"

2) The inputs and the non-IP western blots would be much improved if the phosphorylation of Snf1 at T210 was shown. Cell Signaling Technologies manufactures a suitable rabbit anti phospho AMPK α T172 antibody.

3) The background phosphorylations of Tda1 and Hxk2 in cells grown in 2% glucose could be due to activation of Snf1 by centrifugation prior to killing of cells with 20% TCA (Orlova M et al 2008; PMID: 18949820 DOI: 10.1002/yea.1628). The cells should be killed prior to centrifugation, for example by adding 1ml 100% TCA to 4 ml of media. (Azide will not kill yeast). An alternative method is to boil the yeast, treat with 0.2M NaOH and extract into sample buffer (Orlova M et al 2008). Please could some experiments be repeated with the cells killed before centrifugation to determine if this is the cause of the background phosphorylation.

4) The localization of Hxk2 should be determined in the Tda1, Snf1, and GSK3 mutants (and combinations). Does the enhanced respiration growth and gene expression of *tad1* Δ cells require Hxk2 to be nuclear? More needs to be done here, since these results suggest that a nuclear localization of Hxk2 is pro-respiration.

5) Have any experiments been conducted with phospho-mimicking mutations?

6) The authors should show the Inputs in Figures:

1 F, G, H.

3E, F. Please show the inputs

4C.

6A. Although there is some apparent increase in the growth rate of Hxk2-S15A and *tad1* cells in glycerol compared with WT cells, this experiment would be improved and clearer by monitoring growth of cells in liquid media using a plate reader.

Minor corrections

Line 32: Should read "the Tda1 protein kinase"

Line 56: Lbk1 is a mammalian Snf1 kinase. Please cite Sak1, Elm1, Tos3 as *S. cerevisiae* kinases that phosphorylate Snf1 at T210

Line 58: The authors should also note that Snf1 is also inhibited by SUMOylation as a result of PKA activity in glucose, and that Pma1 activity inhibits Snf1 by cytoplasmic alkalization causing deprotonation of the N-terminal polyHIS motif.

Lines 65-78: This is rather repetitive and could be best merged into previous paragraphs.

Line 157. S483 T484 phosphorylation is NOT essential for Hxk2 phosphorylation, as Hxk2 is still phosphorylated in their absence, albeit to a lesser extent. You can say "important" instead. Ditto for the word "critical" in line 165.

Line 267. Please downgrade the claim. It is not "critical"

Line 295: Should read "glucose repressed genes"

Referee #3:

The study of Nonaka et al. investigates the role of the budding yeast protein kinase Tda1 in metabolic signalling, showing its putative role as an AMPK- and GSK3-dependent negative regulator of glucose starvation response. The manuscript is in general nicely written and easy to follow and potentially interesting for a wide range of researchers in the field of metabolic signalling. However, several improvements are necessary:

1. In the introduction, it is necessary to clearly specify the model organism to which the knowledge relates (e.g. on line 100 and

- 104). Also, LKB1 (line 56) is a mammalian protein, which has to be stated and its name has to be corrected (LKB1, not Lkb1).
2. According to Kassir et al., PMID 17100585, Mck1, Rim11, Ygk3, and Mrk1 are homologs of GSK3-beta. This is, in my opinion, correct in all manuscript sections to avoid confusion.
3. For all immunoblots using anti-Hxk2 antibody: an hxk2 Δ control (to exclude presence of a non-specific band) is necessary! Although the authors show a control immunoblot with hxk2 Δ in Sup.Fig.1B, this result is not applicable because the conditions of the experiments are different (low glucose and phostag gels in the main figures, but log phase cells and standard SDS-PAGE gel in the control figure S1B).
4. In all in vitro phosphorylation assays - to draw any conclusion - it is ABSOLUTELY NECESSARY to use the kinase-dead mutants, e.g. Snf1-KD and Tda1-KD (or a specific kinase inhibitor). Without such a control, the results are inconclusive.
5. For coIP experiments - it is important to show blots where the input signals are not saturated (to assess whether the increase in signal in the IP fraction compared to the negative control is not simply due to a difference in the amount of the protein in the lysate).
6. It would be interesting to know how different growth conditions affect the interaction (coIP) between Tda1 and Rim11 or Snf1, to understand the biological function of this interaction. Also, the author may consider using of diploid cells as Rim11 is a key regulator of meiosis initiation and its function in haploid cells may be impacted.

Referee #1**Major points**

- Introduction, l. 54, the authors state "the gamma subunit detects a drop in the cellular (AMP+ADP)/ATP ratio" but it is the opposite- either a drop in ATP/(AMP+ADP) or an increase in (AMP+ADP)/ATP.

Following the referee's comments, a drop in (AMP+ADP)/ATP was revised to a drop in ATP/(AMP+ADP). We have clarified this point in the revised manuscript (Line 54).

- Could the authors discuss the fact that in the *snf1*Δ strain, there is more phosphorylation of Hxk2 to begin with?

In our co-immunoprecipitation experiments, we observed an interaction between Tda1 and Snf1 even under high-glucose conditions (Fig. 1E). We speculate that Snf1 binding to Tda1 may suppress Tda1 activity in these conditions. Conversely, the loss of *SNF1* partially activates Tda1, leading to increased Hxk2 phosphorylation compared to WT cells, as shown in Fig. 1A, B. We have clarified this point in the revised manuscript (Lines 155–159).

- Could the authors quantify the growth differences shown by drop tests in Fig 6A using growth curves. It is an important aspect of the paper and it would be nice to have robust data at this stage.

We agree that the growth curve strengthens the growth data, so we generated growth curves by measuring OD₆₀₀ over time. These curves more clearly demonstrate that the *tda1*Δ and Hxk2-S15A mutants exhibit enhanced growth compared to WT in YP medium with 3% glycerol, whereas the *snf1*Δ and *rim11*Δ *mck1*Δ mutants show suppressed growth. We have clarified this point in the revised manuscript (Fig. 6B, Lines 287–291).

- Is this restricted to glycerol, or also observed for other C sources such as raffinose or galactose?

Although we did not conduct assays in raffinose or galactose medium, experiments using medium with 2% sodium acetate as a non-fermentable carbon source yielded similar conclusions. We have clarified this point in the revised manuscript (Fig. 6A, Lines 283–287).

- Also, how is the HXK2-S15A generated? Is it a clean replacement at the endogenous locus, how was this made? Please include details in Material and methods section.

The method for constructing the Hxk2-S15A strain is described in the Methods section (Lines 425-444).

- Same question for all the Ser/Thr mutants generated for TDA1.

Following the referee's comment, we have described the method in the same manner as for Hxk2-S15A (Lines 425-444).

- RNA-seq - are the three columns for each mutants biological replicates? What does the colour code correspond to? (What is the reference? WT in Glc medium?). How was clustering made? Are all genes represented? Please provide more information about this experiment in the methods/legend/text sections, as it is it seems a bit expedited.

In response to the referee's comments, we have provided a detailed description of the

RNA-seq data analysis in the Methods, figure legend, and main text as follows. The three columns in the RNA-seq heatmap represent results from three independent experiments for each strain, with the color scale indicating expression levels relative to the average expression of each gene (Lines 845–846). Hierarchical clustering was performed on the top 600 genes using the iDEP v.0.96 web server, and the heatmap was visualized using iDEP v.0.96 (Lines 302, 562-564, 844).

- Fig 6C and 6D. I am not sure about the interpretation of these graphs. Why plot the ratios (xxx/WT) against each other rather than the values themselves? If the values on X and Y axes are divided by the same WT value then it should be the same. What is the coefficient indicated (regression, correlation?) Globally it seems that *tda1*Δ phenocopies Hxk2-S15A and is anti-correlated with *snf1*Δ but I feel that these graphs do not show this so clearly. Would principal component analysis display a better view of the results?

The same plot can be obtained even without normalizing to WT. However, the fact that the ratios of *tda1*Δ/WT and Hxk2-S15A/WT are greater than 1, while that of *snf1*Δ/WT is less than 1 for many genes involved in gluconeogenesis, the TCA cycle, and oxidative phosphorylation suggests that the starvation response is enhanced in *tda1*Δ and Hxk2-S15A mutants but suppressed in *snf1*Δ mutant. This helps to explain why *tda1*Δ and Hxk2-S15A mutants exhibited enhanced growth compared to WT cells, whereas growth was suppressed in the *snf1*Δ mutant under non-fermentable carbon source conditions (Fig. 6A,B). This is consistent with the RT-PCR results, which showed that the expression of glucose-repressed genes was increased in *tda1*Δ and Hxk2-S15A mutants compared to WT cells, while it was suppressed in the *snf1*Δ mutant. (Fig. 6F). We have added an explanation for this (Lines 308-312).

The coefficient indicated correlation. We have added an explanation for this (Line 847).

Following the referee's suggestion, we conducted PCA analysis, which clarified that the gene expression profiles of *tda1*Δ and Hxk2-S15A mutants are similar, while they show a markedly distinct profile compared to the *snf1*Δ mutant. We have clarified this point in the revised manuscript (Appendix Fig. S5 and lines 305-306).

- I was wondering if the authors had tried to look at the growth of a *snf1Δ tda1Δ* or *snf1Δ Hxk2-S15A* on glycerol, to know whether the lack of Hxk2 phosphorylation would restore the expression of glucose-repressed genes in the *snf1Δ*, or on the opposite, whether Snf1 activity is required for the *tda1Δ* or *Hxk2-S15A* phenotypes? These simple experiments could help pinpoint whether *tda1Δ* mimics a glucose derepressed mutant like *reg1Δ*.

It was crucial to determine whether the absence of Hxk2 phosphorylation could restore the expression of glucose-repressed genes in the *snf1Δ* mutant or, conversely, whether Snf1 activity is required for the *tda1Δ* and *Hxk2-S15A* phenotypes. To investigate this, we generated *snf1Δ tda1Δ* and *snf1Δ Hxk2-S15A* mutants and conducted spot assays. We found that these strains exhibited significantly slower growth on glycerol plates compared to the *tda1Δ* and *Hxk2-S15A* mutants, suggesting that the enhanced growth of *tda1Δ* and *Hxk2-S15A* mutants likely requires Snf1 activity. We have clarified this point in the revised manuscript (Appendix Fig. S4 and lines 291-298).

- I think the discussion is a bit long, it could be edited to be more straight-to-the-point and avoid redundancies, but also completed. What do we know about the relationships between glucose signaling / metabolism and Rim11/Mck1 (echoing to the data presented in 6A which suggests a lack of glucose repression?). How would the glucose signal be perceived by GSK-3, and/or what could be the mechanism by which Hxk2 becomes suddenly phosphorylated by GSK-3? (eg. the priming site Thr513, for example). Overall, can the authors put forward hypotheses on how Hxk2 phosphorylation controls gene expression? For example data in the literature shows that Hxk2 monomers and dimers have a different affinity towards glucose. Also, data by Lesko et al 2023 (doi:10.1371/journal.pgen.1010745) suggest that *hxx2Δ* does not confer major transcriptional changes. How the authors reconcile these data with their data on the *Hxk2-SA* mutant? This should be discussed. Finally, I think there is a paradox that is not discussed. Snf1 is a positive regulator of Tda1 activity, yet the transcriptomics profile are opposite to each other. Maybe this could be

mentioned and discussed.

Accoering to the referee's comment, we have revised the Discussion section.

- What do we know about the relationships between glucose signaling / metabolism and Rim11/Mck1 (echoing to the data presented in 6A which suggests a lack of glucose repression?). How would the glucose signal be perceived by GSK-3, and/or what could be the mechanism by which Hxk2 becomes suddenly phosphorylated by GSK-3? (eg. the priming site Thr513, for example).

It is known that Rim11 and Mck1 are inhibited by PKA. Under glucose starvation conditions, PKA activity is suppressed, allowing yeast GSK3- β to become activated. Since the details of Thr513 phosphorylation are unclear, it remains uncertain whether Thr513 acts as a trigger that enables GSK3- β to phosphorylate Tda1. We have addressed this point in the Discussion section. (Lines 357-370).

- Overall, can the authors put forward hypotheses on how Hxk2 phosphorylation controls gene expression? For example data in the literature shows that Hxk2 monomers and dimers have a different affinity towards glucose. Also, data by Lesko et al 2023 (doi:10.1371/journal.pgen.1010745) suggest that *hvk2* Δ does not confer major transcriptional changes. How the authors reconcile these data with their data on the Hxk2-SA mutant? This should be discussed.

Since Hxk2-S15A mutant retains hexokinase activity, its glycolytic efficiency differs from that of the *hvk2* Δ mutant, likely resulting in changes in transcriptional profiles. Further detailed studies are needed to investigate this point. We have addressed this point in the Discussion section (Lines 391-399).

-Finally, I think there is a paradox that is not discussed. Snf1 is a positive regulator of Tda1 activity, yet the transcriptomics profile are opposite to each other. Maybe this could be mentioned and discussed.

Based on the phenotype of the *TDA1* deletion mutant, we hypothesize that Tda1 functions as a brake on gene expression. On the other hand, Snf1 activates not only Tda1 but also various other transcriptional activators, which likely leads to the observed differences in the impact on the transcriptome between Snf1 and Tda1. We propose that

Tda1 activated by Snf1 fine-tunes gene expression in parallel with other transcription factors. This point has been elaborated further in the Discussion section (Lines 371-381).

Minor points.

- the authors state "when glucose levels drop, more efficient energy production pathways are activated" (l.71) - I'm not sure about the word "efficient" here, yeast cells indeed trigger the onset of respiration but they rather turn to other carbon sources. In a way it is more efficient because it is better adapted, but "efficient" could also be read as a better output in ATP and I'm not sure this is the case

In response to the referee's comment, we have revised "when glucose levels drop, more efficient energy production pathways are activated" to "when glucose levels drop, alternative metabolic pathways are activated" (Lines 62-63).

- The authors refer to past studies by the Moreno lab on the potential nuclear functions of Hxk2. This work has been criticized (Kriegel et al. 2016; doi:10.1074/jbc.L116.735514) and invalidated (Lesko et al 2023; doi:10.1371/journal.pgen.1010745). Same thing for the data suggesting that Snf1 is the Hxk2 kinase (Fernandez-Garcia et al., 2012, doi: 10.1074/jbc.M112.401679), which was not confirmed in subsequent studies (Kaps et al. 2015; doi:10.1074/jbc.M114.595074). While I agree that these studies could be mentioned, I think that now is the time to start downplaying the importance of this work to clear up the field, by being even more cautious about these conclusions when mentioned in the text and/or limit citations. But of course this is at the author's discretion.

The nuclear function of Hxk2 was important for considering the physiological role of Tda1 in this study, which is why we cited it. In the report by Kaps et al., Hxk2 phosphorylation was not completely abolished in the *snf1Δ* strain. However, they did not demonstrate the absence of Hxk2 phosphorylation activity by Snf1 in vitro, so we re-examined this point.

- I believe the authors should mention Mig1 in the introduction (at the same time as Adr1 etc, l. 77)

Following the referee's comments, we included a mention of Mig1 in the introduction (Lines 72-73).

- l.103, "GSK-3 is constitutively active but can be inactivated" - then I suppose that it is not constitutively active? Do the authors mean that it is always active in riche medium?

As the referee pointed out, GSK-3 is not constitutively active. We have revised " GSK-3 is constitutively active but can be inactivated by phosphorylation through upstream kinases like Akt and PKA " to "GSK3- β is inhibited by Akt and PKA, both of which are active under high-glucose conditions. Consequently, similar to AMPK, GSK3- β can be activated during glucose starvation when Akt and PKA are inactivated" (Lines 98-99).

-Typo in Fig 6D, "TCA cyclels"

As the referee pointed out, it was revised to TCA cycles (Fig. 6E).

Referee #2

General comments-Major

1) The title would be clearer if it were: " Independent activation of Tda1 by Snf1 and GSK3 negatively regulates glucose starvation signaling in *S. cerevisiae*"

Due to the character limit (The total length of the title should not exceed 100 characters including spaces), we keep the title as "Tda1, activated by Snf1 and yeast GSK3- β , negatively regulates the glucose starvation signaling".

2) The inputs and the non-IP western blots would be much improved if the phosphorylation of Snf1 at T210 was shown. Cell Signaling Technologies manufactures a suitable rabbit anti phospho AMPK α T172 antibody.

Detecting Snf1 phosphorylation at T210 using a rabbit anti-phospho-AMPK α T172 antibody could serve as an indicator of Snf1 activation under low-glucose conditions. However, our primary focus is to clarify the regulatory mechanism of Tda1 and its role under glucose starvation conditions, and given that numerous previous studies have established Snf1 activation under low-glucose conditions, we did not assess Snf1 phosphorylation at T210 in this study.

3) The background phosphorylations of Tda1 and Hxk2 in cells grown in 2% glucose could be due to activation of Snf1 by centrifugation prior to killing of cells with 20% TCA (Orlova M et al 2008; PMID: 18949820 DOI: 10.1002/yea.1628). The Cells should be killed prior to centrifugation, for example by adding 1ml 100% TCA to 4 ml of media. (Azide will not kill yeast). An alternative method is to boil the yeast, treat with 0.2M NaOH and extract into sample buffer (Orlova M et al 2008). Please could some experiments be repeated with the cells killed before centrifugation to determine if this is the cause of the background phosphorylation.

Thank you for the referee's careful feedback. To prevent background phosphorylation, we immediately inactivated cellular processes by adding TCA directly to the medium at a final concentration of 10% at the time of collection. This procedure is described in the Methods section (Lines 458–459). Even under these conditions, we still detected faint phosphorylation of Hxk2 in WT cells under high-glucose conditions (Fig. 1A). We speculate that this may be due to slight activation of Tda1 by Snf1 and GSK3- β , as no Hxk2 phosphorylation was observed in *tda1* Δ and *snf1* Δ , *rim11* Δ *mck1* Δ mutants under high-glucose conditions (Figs. 1A, 4A).

4) The localization of Hxk2 should be determined in the Tda1, Snf1, and GSK3 mutants (and combinations). Does the enhanced respiration growth and gene expression of *tda1* Δ cells require Hxk2 to be nuclear? More needs to be done here, since these results suggest that a nuclear localization of Hxk2 is pro-respiration.

A key question was whether the enhanced growth and gene expression observed in *tda1* Δ cells on a non-fermentable carbon source depend on the nuclear localization of Hxk2. To investigate this, we examined the subcellular localization of endogenous Hxk2 using immunostaining with an anti-Hxk2 antibody in WT, *tda1* Δ , Hxk2-S15A, *snf1* Δ , *rim11* Δ *mck1* Δ , and *snf1* Δ *rim11* Δ *mck1* Δ mutants (see figure below).

As previously reported by Lesko et al., a fraction of Hxk2 localizes to the nucleus under glucose starvation in WT cells. However, we observed no differences in Hxk2 localization between WT and the tested mutant strains. These findings suggest that Hxk2 phosphorylation at Ser15 and the deletion of upstream kinases do not influence its nuclear localization.

The connection between Hxk2-mediated gene expression and its nuclear localization remains an intriguing avenue for future research. Since our study primarily focuses on the regulatory mechanism of Tda1 and its role under glucose starvation, we have chosen not to include these results in this manuscript. Additionally, a recent in-depth study by Lesko et al. provides extensive insights into Hxk2 localization. As our findings do not offer significantly new information on this topic, we believe their exclusion is justified.

Figure for referee with unpublished data and its description has been removed upon request by the authors.

5) Have any experiments been conducted with phospho-mimicking mutations?

We have not conducted experiments using phospho-mimicking mutants, as it is widely recognized that these mutations often fail to accurately mimic protein phosphorylation.

6) The authors should show the Inputs in Figures:

1 F, G, H.

3E, F. Please show the inputs

4C.

Following the referee's comment, we provided the Inputs (Figs. 1F,G,H,3E,F and 4C).

6A. Although there is some apparent increase in the growth rate of Hxk2-S15A and tda1D cells in glycerol compared with WT cells, this experiment would be improved and clearer by monitoring growth of cells in liquid media using a plate reader.

We agreed that the growth curve strengthens the growth data, so we created growth curves by measuring OD600 over time. The growth curves more clearly show that the *tda1*Δ and Hxk2-S15A mutants exhibit enhanced growth compared to WT in YP medium with 3% glycerol, while the *snf1*Δ and *rim11*Δ *mck1*Δ mutants show suppressed growth. We have clarified this point in the revised manuscript (Fig. 6B and lines 287-291).

Minor corrections

Line 32: Should read "the Tda1 protein kinase"

Following the referee's comment, "kinase protein Tda1" was revised to "Tda1 protein

kinase” (Line 32)

Line 56: Lbk1 is a mammalian Snf1 kinase. Please cite Sak1, Elm1, Tos3 as S. cerevisiae kinases that phosphorylate Snf1 at T210

This point was cited in Introduction (Line 66).

Line 58: The authors should also note that Snf1 is also inhibited by SUMOylation as a result of PKA activity in glucose, and that Pma1 activity inhibits Snf1 by cytoplasmic alkalization causing deprotonation of the N-terminal polyHIS motif.

Following the referee’s comment, we have added a description stating that Snf1 is inhibited by SUMOylation and deprotonation of the polyHis motif in Introduction section (Lines 66-69).

Lines 65-78: This is rather repetitive and could be best merged into previous paragraphs.

Following the referee’s comment, we merged the previous paragraph into a single unified paragraph (Lines 47-74).

Line 157. S483 T484 phosphorylation is NOT essential for Hxk2 phosphorylation, as Hxk2 is still phosphorylated in their absence, albeit to a lesser extent. You can say "important" instead. Ditto for the word "critical" in line 165.

Following the referee’s comment, we have replaced "essential" in Line 154 with "important" (Line 160) and "critical" in Line 165 with "crucial" (Line 169).

Line 267. Please downgrade the claim. It is not "critical"

Following the referee's comment, we have replaced "critical" in Line 264 with "crucial" (Line 272).

Line 295: Should read "glucose repressed genes"

Following the referee's comment, " glucose starvation response " was revised to " glucose repressed genes " (Line 322).

Referee #3

1. In the introduction, it is necessary to clearly specify the model organism to which the knowledge relates (e.g. on line 100 and 104). Also, LKB1 (line 56) is a mammalian protein, which has to be stated and its name has to be corrected (LKB1, not Lkb1).

I have clarified the expression in Line 100 (Line 97). In Line 104, I refrained from specifying examples because it has been reported in numerous eukaryotes to an extent that makes enumeration impractical (Line 102).

2. According to Kassir et al., PMID 17100585, Mck1, Rim11, Ygk3, and Mrk1 are homologs of GSK3-beta. This is, in my opinion, correct in all manuscript sections to avoid confusion.

In accordance with the referee's comments, the notation was standardized to GSK3- β instead of GSK-3.

3. For all immunoblots using anti-Hxk2 antibody: an *hxk2* Δ control (to exclude presence of a non-specific band) is necessary! Although the authors show a control immunoblot with *hxk2* Δ in Sup.Fig.1B, this result is not applicable because the conditions of the experiments are different (low glucose and phostag gels in the main figures, but log phase cells and standard SDS-PAGE gel in the control figure S1B).

To evaluate the specificity of the anti-Hxk2 antibody, we conducted low-glucose chase experiments in WT and *hxk2* Δ cells. Total cell lysates were separated by Phos-tag SDS-PAGE (to detect Hxk2) or standard SDS-PAGE (to detect Cdc48) and analyzed by immunoblotting with anti-Hxk2 and anti-Cdc48 antibodies. The Hxk2 band was detected in WT lysates but was absent in *hxk2* Δ , confirming that the anti-Hxk2 antibody specifically recognizes Hxk2 (Appendix Fig. S1B).

4. In all in vitro phosphorylation assays - to draw any conclusion - it is ABSOLUTELY NECESSARY to use the kinase-dead mutants, e.g. Snf1-KD and Tda1-KD (or a specific kinase inhibitor). Without such a control, the results are inconclusive.

Since we agreed that a definitive conclusion could not be reached without using kinase-dead strains, we generated kinase-dead mutants of Tda1, Snf1, and Rim11 and conducted in vitro phosphorylation assays using these mutants. As expected, the kinase-dead mutants were unable to promote the phosphorylation of their target proteins. We have clarified this point in the revised manuscript (Figures 1F, H, 3E and Lines 144, 148-149, 197-198).

5. For coIP experiments - it is important to show blots where the input signals are not saturated (to assess whether the increase in signal in the IP fraction compared to the negative control is not simply due to a difference in the amount of the protein in the lysate).

Following the referee's comment, we included images where the input signals are not saturated (Figs. 1E and 3A)

6. It would be interesting to know how different growth conditions affect the interaction (coIP) between Tda1 and Rim11 or Snf1, to understand the biological function of this interaction.

Since the interaction between Tda1 and Snf1 or Rim11 under different growth conditions was of interest, we performed co-immunoprecipitation (Co-IP) assays under both high- and low-glucose conditions. Our results showed that Tda1 interacts with Snf1 and Rim11 in both conditions (Figs. 1E, 3A). We speculate that under low-glucose conditions, Snf1 or Rim11 binding promotes Tda1 phosphorylation, whereas under high-glucose conditions, this interaction may modulate, or potentially suppress, Tda1 activity.

Also, the author may consider using of diploid cells as Rim11 is a key regulator of meiosis initiation and its function in haploid cells may be impacted.

Since this study focuses on the regulation of Tda1 by glucose availability, experiments using diploid cells were beyond the scope of our current investigation.

Dear Prof. Kamura,

Thank you for submitting your revised manuscript. It has now been seen by two of the original referees.

As you can see, referees find that the study is significantly improved during revision and recommend publication. However, I need you to address the points below before I can accept the manuscript.

- Please remove the 'Author Contributions Statements' section from the manuscript text.
- We note that there are 6 Appendix Tables uploaded separately.
 - o We find that Appendix Table 2 and 3 are too large and therefore it should be converted to a Dataset and should be renamed as Dataset EV1 and Dataset EV2 respectively (file names and in-text callouts should be updated accordingly).
 - o We find that Appendix Tables 4 and 5 should be a part of the Reagents & Tools table to increase the accessibility of the information.
 - o The remaining Appendix Tables (1 and 6) should be included in the Appendix pdf.
- The source data need to be reorganized - the source data of the main figures should be uploaded as one zip file per figure. The rest of the source data can stay as a single zip file.
- Materials and Methods section needs to be renamed as Methods.
- Please make the datasets GSE290495 and PXID: PXD060707 publicly available and remove the reviewer's tokens from the manuscript.
- Our production/data editors have asked you to clarify several points in the figure legends:
 - o Please note that the legends for figures 3 is not provided in the sequential manner (legend for figure 3G, J is provided before legend of figure 3C). This needs to be rectified.
 - o - Please note that the legends for figures 4 is not provided in the sequential manner (legend for figure 3D, F is provided before legend of figure 4B). This needs to be rectified.
 - o - Please note that the exact p values are not provided in the legends of figures 1B, C; 2B, 3C, D, H, I, K, L; 4B, E, G; 5B, D; 6B, F
 - o - Please note that the measure of center for the error bars needs to be defined in the legends of figures 1B, C; 2B, 3C, D, H, I, K, L; 4B, E, G; 5B, D; 6B, F

Thank you again for giving us to consider your manuscript for EMBO Reports, I look forward to your minor revision.

Kind regards,

Deniz Senyilmaz Tiebe

--

Deniz Senyilmaz Tiebe, PhD
Senior Scientific Editor
EMBO Reports

Referee #1:

I thank the authors for their consideration. The authors have satisfactorily addressed my comments and the data provided support their model so I now recommend publication.

Referee #2:

The authors have successfully addressed all of my criticisms, the paper can be published in its current form.

- Please remove the 'Author Contributions Statements' section from the manuscript text.

Following the comment, the Author Contributions section has been removed.

- We note that there are 6 Appendix Tables uploaded separately.
 - We find that Appendix Table 2 and 3 are too large and therefore it should be converted to a Dataset and should be renamed as Dataset EV1 and Dataset EV2 respectively (file names and in-text callouts should be updated accordingly).

Following the comment, Appendix Table 2 and 3 have been converted to Dataset EV1 and Dataset EV2, respectively. The file names and in-text citations have also been updated accordingly (Lines 208, 306).

- We find that Appendix Tables 4 and 5 should be a part of the Reagents & Tools table to increase the accessibility of the information.

As per the comment, the contents of Appendix Tables 4 and 5 have been added to the Reagents & Tools table.

- The remaining Appendix Tables (1 and 6) should be included in the Appendix pdf.

As per the comment, Appendix Tables 1 and 6 have been combined into a single file. Appendix Table 6 has been renamed to Appendix Table 2.

- The source data need to be reorganized - the source data of the main figures should be uploaded as one zip file per figure. The rest of the source data can stay as a single zip file.

Following the comment, the source data of the main figures have been organized into separate zip files, one for each figure.

- Materials and Methods section needs to be renamed as Methods.

According to the comment, the Materials and Methods section has been renamed to Methods (Line 414).

- Please make the datasets GSE290495 and PXID: PXD060707 publicly available and remove the reviewer's tokens from the manuscript.

According to the comment, we have made the datasets GSE290495 and PXID: PXD060707 publicly available and have removed the reviewer's tokens from the manuscript.

- Our production/data editors have asked you to clarify several points in the figure legends:
 - o Please note that the legends for figures 3 is not provided in the sequential manner (legend for figure 3G, J is provided before legend of figure 3C). This needs to be rectified.

Following the comment, the legend for Figure 3 has been revised to follow a sequential manner (Lines 786-830).

- o - Please note that the legends for figures 4 is not provided in the sequential manner (legend for figure 3D, F is provided before legend of figure 4B). This needs to be rectified.

Following the comment, the legend for Figure 4 has been revised to follow a sequential manner (835-873).

o - Please note that the exact p values are not provided in the legends of figures 1B, C; 2B, 3C, D, H, I, K, L; 4B, E, G; 5B, D; 6B, F

Following the comment, exact p-values have been added to the legends of figures 1B, C; 2B; 3C, D, H, I, K, L; 4B, E, G; 5B, D; 6B, F (Lines 742-746, 750-752, 775-779, 788-793, 796-799, 809-813, 816-818, 822-825, 827-830, 838-845, 853-861, 865-873, 881-885, 889-897, 908-925, 938-947). However, GraphPad Prism did not calculate exact values for $p < 0.0001$.

o - Please note that the measure of center for the error bars needs to be defined in the legends of figures 1B, C; 2B, 3C, D, H, I, K, L; 4B, E, G; 5B, D; 6B, F

As per the comment, it has been specified that the error bars represent mean \pm SD (Lines 740-741, 748-749, 774, 787, 794-795, 807-808, 814-815, 820-821, 826-827, 836, 852, 863-864, 879, 887-888, 906-907, 936-937).

Prof. Takumi Kamura
Nagoya University
Graduate School of Science
Furo-cho Chikusa-ku
Nagoya City, Aichi 464-8602
Japan

Dear Prof. Kamura,

Thank you for submitting your revised manuscript. I have now looked at everything and all is fine. Therefore, I am very pleased to accept your manuscript for publication in EMBO Reports.

Congratulations on a nice work!

Before we can transfer your manuscript to our production team, I need your input on one more point. I performed some minor changes to increase clarity and accessibility of the findings. Please confirm, or feel free to propose further changes. Thank you.

Title: Snf1 and yeast GSK3- β activates Tda1 to suppress glucose starvation signaling

Abstract: In budding yeast, the presence of glucose, a preferred energy source, suppresses the expression of respiration-related genes through a process known as glucose repression. Conversely, under glucose starvation conditions, Snf1 phosphorylates and activates downstream factors, relieving this repression and allowing cells to adapt. Recently, the Tda1 protein kinase has been implicated in these glucose starvation responses, although its function remains largely uncharacterized. In this study, we demonstrate that Snf1 and yeast glycogen synthase kinase 3-beta (GSK3- β) independently phosphorylate and activate Tda1, which in turn phosphorylates Hxk2 at Ser15. The Ser483 and Thr484 residues of Tda1 are critical for its activation by Snf1, while the Ser509 residue is crucial for its activation by yeast GSK3- β . Importantly, under glucose starvation conditions, the TDA1 deletion mutant shows increased expression of respiration-related genes and a faster growth rate compared to wild-type cells, which is opposite to what is observed in SNF1 and yeast GSK3- β deletion mutants. These findings suggest that Tda1 is activated by Snf1 and yeast GSK3- β , and functions as a suppressor of the glucose starvation signaling.

Kind regards,

Deniz Senyilmaz Tiebe

--

Deniz Senyilmaz Tiebe, PhD
Senior Scientific Editor
EMBO Reports
